# Activation and substrate specificity of the human P4-ATPase ATP8B1

Thibaud Dieudonné[1,7] ✉, Felix Kümmerer[2], Michelle Juknaviciute Laursen[1], Charlott Stock [1], Rasmus Kock Flygaard[1], Syma Khalid [3], Guillaume Lenoir [4], Joseph A. Lyons [5,6], Kresten Lindorff-Larsen[2] & Poul Nissen [1] ✉

Asymmetric distribution of phospholipids in eukaryotic membranes is essential for cell integrity, signaling pathways, and vesicular trafficking. P4-ATPases, also known as flippases, participate in creating and maintaining this asymmetry through active transport of phospholipids from the exoplasmic to the cytosolic leaflet. Here, we present a total of nine cryo-electron microscopy structures of the human flippase ATP8B1-CDC50A complex at 2.4 to 3.1 Å overall resolution, along with functional and computational studies, addressing the autophosphorylation steps from ATP, substrate recognition and occlusion, as well as a phosphoinositide binding site. We find that the P4-ATPase transport site is occupied by water upon phosphorylation from ATP. Additionally, we identify two different autoinhibited states, a closed and an outward-open conformation. Furthermore, we identify and characterize the $PI(3,4,5)P_3$ binding site of ATP8B1 in an electropositive pocket between transmembrane segments 5, 7, 8, and 10. Our study also highlights the structural basis of a broad lipid specificity of ATP8B1 and adds phosphatidylinositol as a transport substrate for ATP8B1. We report a critical role of the sn-2 ester bond of glycerophospholipids in substrate recognition by ATP8B1 through conserved S403. These findings provide fundamental insights into ATP8B1 catalytic cycle and regulation, and substrate recognition in P4-ATPases.

Lipid flippases actively transport lipids from the exoplasmic leaflet to the cytosolic leaflet of cellular membranes thereby creating and maintaining an asymmetric lipid distribution between the two leaflets[1]. Transbilayer lipid asymmetry in the late secretory pathway is critical for eukaryotic cell functions such as membrane homeostasis, migration, cell signaling, blood coagulation, and morphogenesis[2]. Most eukaryotic lipid flippases belong to the P4-ATPases subfamily of the P-type ATPase superfamily (with the sole known exception of ABCA4)[3,4]. The P-type ATPases share a common architecture with three

cytosolic domains that are involved in the autocatalyzed ATP hydrolysis: the nucleotide-binding (N) domain binds ATP, the phosphorylation (P) domain contains a conserved aspartate residue, which is phosphorylated during the transport cycle, and the actuator (A) domain contains a conserved glutamate, which is responsible for the subsequent dephosphorylation of the phosphoenzyme intermediate, that enables the protein to return to its initial state[5] (Supplementary Fig. 1a). These three domains are linked to the membrane domain, which is typically formed by ten transmembrane segments (TM1–10).

[1]DANDRITE, Nordic EMBL Partnership for Molecular Medicine, Department of Molecular Biology and Genetics, Aarhus University, Aarhus, Denmark. [2]Structural Biology and NMR Laboratory & Linderstrøm-Lang Centre for Protein Science, Department of Biology, University of Copenhagen, Copenhagen, Denmark. [3]Department of Biochemistry, University of Oxford, Oxford, UK. [4]Université Paris-Saclay, CEA, CNRS, Institute for Integrative Biology of the Cell (I2BC), 91198 Gif-sur-Yvette, France. [5]Department of Molecular Biology and Genetics, Aarhus University, Aarhus, Denmark. [6]Interdisciplinary Nanoscience Centre (iNANO) Aarhus University, Aarhus, Denmark. [7]Present address: Université Paris-Saclay, CEA, CNRS, Institute for Integrative Biology of the Cell (I2BC), 91198 Gif-sur-Yvette, France. ✉e-mail: thibaud.dieudonne@i2bc.paris-saclay.fr; pn@mbg.au.dk

TM4 contains a conserved, proline-containing motif, which is specific to each P-type subgroup. For P4-ATPases it is P(I/V)S(L/M) and it constitutes part of the transport binding site and forms a hydrophobic gate preventing lipids from diffusing freely through the protein[6,7] (Supplementary Fig. 1a).

Substrate transport by P-type ATPases is driven by large conformational changes of the transporter switching between so-called E1 and E2 states that allow alternating access to each side of the membrane. These conformational changes are tightly coupled to the phosphorylation and dephosphorylation reactions performed by the cytosolic domains. The different steps of the P-type ATPase catalytic cycle are described by the Post-Albers scheme[8,9] (Supplementary Fig. 1b), with the transition from the E1 state to E1P state being the phosphorylation reaction, and the transition from the E2P state to E2 state being dephosphorylation. In P4-ATPases, substrate lipid recognition occurs in the outward-open E2P state followed by cognate substrate-induced occlusion triggering dephosphorylation[10–12]. In addition, most P4-ATPases form a binary complex with a CDC50 subunit[3], and this interaction is essential for the correct subcellular membrane targeting and flippase activity of the complex[13–15].

The human genome encodes 14 different P4-ATPases from ATP8, ATP9, ATP10, and ATP11 groups, each containing different isoforms[3]. ATP8B1 was originally identified as the causative gene for progressive familial intrahepatic cholestasis 1 (PFIC1), a rare but severe liver disease characterized by impaired bile flow[16]. Milder diseases caused by mutations of the *ATP8B1* gene include benign recurrent intrahepatic cholestasis of type 1 (BRIC1) and intrahepatic cholestasis of pregnancy (ICP1). The function of ATP8B1 has been associated with the integrity of the canalicular membrane of hepatocytes[17], but the exact mechanism is still unclear, in part obscured by discrepancies regarding the lipid substrate specificity. Furthermore, genetic variation at a site upstream of the ATP8B1 coding region has been identified in resilience to Alzheimer's Disease[18], and loss-of-function mutations of the closely related ATP8B4 represent Alzheimer's risk[19], but in both cases again through unknown mechanisms.

Initially, ATP8B1 was reported as a phosphatidylserine (PS)-specific flippase, but other studies have since proposed specificity to phosphatidylethanolamine (PE), phosphatidylcholine (PC) or cardiolipin (CL)[20–25]. More recently, ATPase activity measurements using the purified ATP8B1-CDC50A and ATP8B1-CDC50B complexes also lead to contrasting conclusions[26,27]. Previously, we reported that PC, PE and PS were able to stimulate the ATPase activity, but not CL or sphingomyelin (SM)[26], whereas Cheng et al., found that PS, but not PC could stimulate activity[27]. Additionally, ATP8B1 activity is tightly regulated by its N- and C-terminal ends that closely interact with the cytosolic domains in an autoinhibited E2P state[26,27]. Furthermore, ATP8B1 activity depends not only on a substrate lipid, but also on the presence of phosphoinositides, with a strong affinity for the triple-phosphorylated form, PI(3,4,5)P$_3$[26].

Here, we present nine different cryo-EM structures, functional studies, and molecular dynamics simulation data of the ATP8B1-CDC50A complex in conformations covering most of the transport cycle and different substrate complexes. From our data, we identify the presence of a water molecule associated with the E1 states of the cycle and located in the canonical transport site of P-type ATPase. We also report that the autoinhibited ATP8B1-CDC50A complex fluctuates between an outward-open and a closed conformation, identify a PI(3,4,5)P$_3$ binding site, new transport substrates, and reveal structural determinants of the broad glycerophospholipid specificity of ATP8B1.

## Results

### Cryo-EM structures of the ATP8B1-CDC50A complex along its catalytic cycle
We expressed the human ATP8B1-CDC50A complex in *Saccharomyces cerevisiae*. Isolated membranes were solubilized with n-Dodecyl β-D-maltoside (DDM) and purified by affinity purification via a BAD-tag (biotin acceptor domain)[28]. DDM was exchanged for Lauryl Maltose Neopentyl Glycol (LMNG) prior to further purification by size-exclusion chromatography. We used two different constructs for our studies: full-length ATP8B1 (WT-ATP8B1) and a variant containing an engineered 3C protease site allowing autoinhibition of ATP8B1 to be relieved by proteolytic cleavage of the C-terminal end (ΔCter-ATP8B1) (Supplementary Fig. 2). The ΔCter-ATP8B1 construct was incubated with nucleotides (ADP or the non-hydrolyzable ATP analog adenine-(βγ-methylene)-triphosphate (AMPPCP)) and/or AlF$_x$ to stabilize forms similar to E1-ATP, E1P-ADP and E1P conformations, corresponding to autophosphorylation from ATP and subsequent ADP release events (Fig. 1 and Supplementary Figs. 3–16, Supplementary Table 1). Furthermore, the phosphate mimics BeF$_x$ and VO$_3^-$ (metavanadate) were used in presence of transport substrate to stabilize the E2P outward-open and the E2···Pi lipid-occluded conformation of the complex (Fig. 1 and Supplementary Figs. 3–16, Supplementary Table 1). Finally, the full-length ATP8B1 preincubated with ATP revealed two off-cycle autoinhibited states (Fig. 1 and Supplementary Figs. 3–16).

### A equivalent binding site of P-type ATPases is occupied by water in the E1 half-cycle of ATP8B1
The cryo-EM structure of the ATP8B1-CDC50A complex in the presence of AMPPCP mimics an ATP-bound E1 state and reveals the ATP binding pocket of ATP8B1 (Fig. 2a, b). As expected, AMPPCP is bound similarly to other P-type ATPases. The adenosine base is stacked by a π-π-interaction with F596. Additionally, D554 is directly engaged in the coordination network of the ß-phosphate, mediated by R652. Indeed, the R652 rotamer state is constrained by interactions with D554 and D734 (Fig. 2b). Noteworthy, the D554N mutation has been identified in patients suffering from PFIC1[29]. Interestingly, extra density was also observed at an interacting distance to the ß-phosphate of AMPPCP, suggesting the presence of an additional cation or a water molecule (see Supplementary Discussion). The E1P-ADP state mimicked by ADP and AlF$_x$ is similar to previously published structures of P4-ATPases[7,12,30]. However, closer inspection of all E1 states along with the outward-open E2P states (autoinhibited and active conformations, see further below) reveals the presence of a density at the canonical, phosphorylation-coupled transport site of P-type ATPases between TM4, TM5 and TM6 (Fig. 2c). Due to the geometry of the site with no more than four ligands, we modeled the site as a water molecule. Interestingly, the water molecule is at interacting distance to S403 from the PISL motif of TM4, the backbone carbonyl group of two residues from TM6 (N989 and T993) and the highly conserved K957 of TM5, which is also described as essential in ATP11C[30].

Given the high conservation of these residues among all P4-ATPases, we compared the cryo-EM densities of previously published P4-ATPases obtained at a sufficient resolution to observe a coordinated water molecule (Supplementary Fig. 17). In almost all maps, a density could be observed at this site, consistent with a water molecule. The water molecule site partially overlaps with transition metal and cation sites of P1 and P2-ATPases (Supplementary Fig. 18). Furthermore, the recent cryo-EM structure of human ATP13A2 (a P5-ATPase), in the E1-ATP conformation also revealed water molecules in the canonical P-type ATPase binding site[31] (Supplementary Fig. 18).

### Autoinhibited ATP8B1-CDC50A can adopt an open and a closed conformation
Previously, two independent structures of the ATP8B1-CDC50A complex in the E2P$_{autoinhibited}$ state were reported[26,27]. Surprisingly, they are not identical, especially in the N-terminal part of the transmembrane region (Supplementary Fig. 19a), where TM1 and TM2 have a different orientation either in a "close" or "open" conformation. Here, we collected cryo-EM data of the full-length ATP8B1-CDC50A complex preincubated with ATP and the activating lipid PI(3,4,5)P$_3$. Upon data

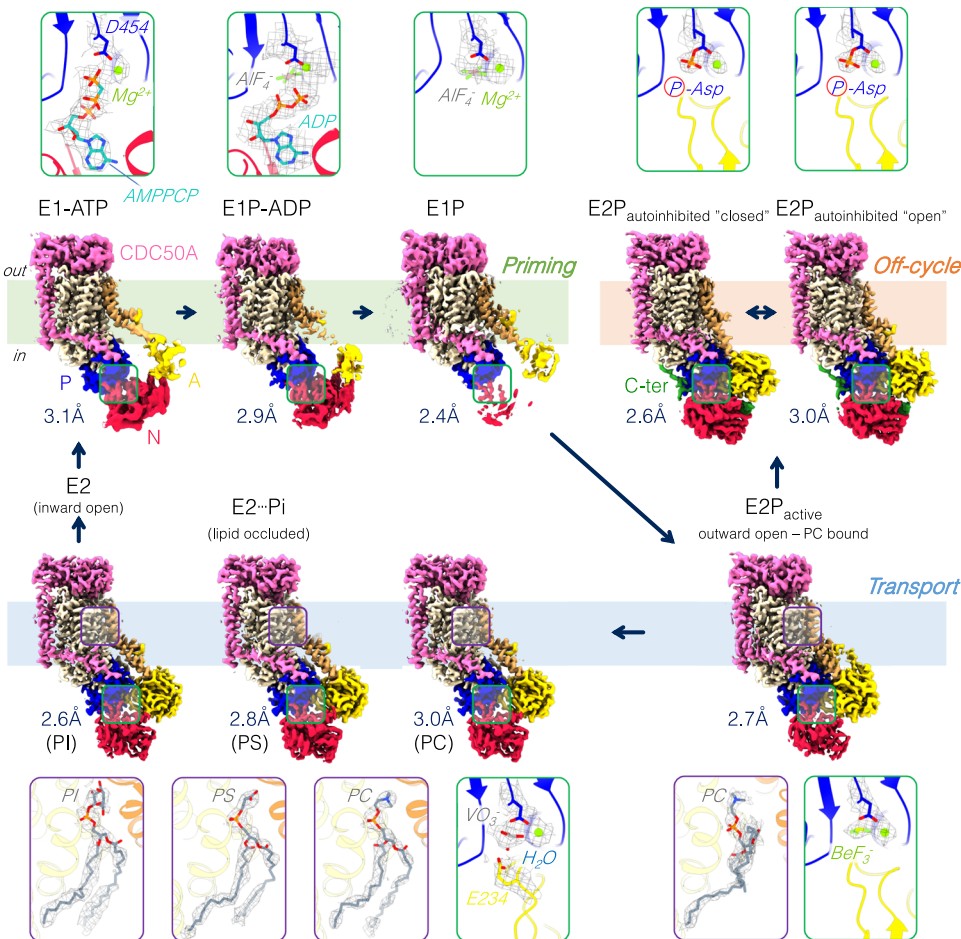

**Fig. 1 | Cryo-EM maps of the ATP8B1-CDC50A flippase complex.** Cryo-EM maps of ATP8B1 in complex with CDC50A in different conformations: E1-ATP, E1P-ADP, E1P, E2P$_{autoinhibited}$ "closed", E2P$_{autoinhibited}$ "open", E2P-active, E2∙∙Pi (PC), E2∙∙Pi (PS) and E2∙∙Pi (PI). For each conformation, a close-up view of the phosphorylation site (green frame) and/or of the lipid transport site (purple frame) including their associated EM maps (black mesh) is shown. "P-Asp", circled in red, refers to the aspartyl-phosphoanhydride residue formed after phosphorylation of the D454 aspartate from ATP. Color code: The cytosolic A-, N-, and P-domains of ATP8B1 are colored in yellow, red, and blue, respectively. The transmembrane domain of ATP8B1 is colored in wheat, with TM1 and TM2 in brown. The C-terminal tail of ATP8B1 is colored in green. CDC50A is colored in pink.

processing two different conformations were observed (Fig. 3). The first one, representing ~90% of the particles, was in a "closed" conformation virtually identical to previous studies of autoinhibited forms[7,26,32] (Figs. 1 and 3 and Supplementary Fig. 19b). In this conformation, TM1 and TM2 are tightly packed to the rest of the transmembrane helix bundle, thus masking the substrate recognition PISL motif of TM4 and closing the exoplasmic lipid entry (Fig. 3 and Supplementary Fig. 19b). However, approx. 10% of the particles adopted an "open" conformation corresponding to the structure reported by Cheng et al.,[27]. Here, TM1-2 are rotated and the lipid-binding groove is open as observed for non-autoinhibited P4-ATPases[30,32,33] (Fig. 3, Supplementary Fig. 20). In both cases, we still observe clear densities for the N- and C-terminal tails, interacting with the A, P and N domains in a similar manner (Fig. 3).

To better understand whether the TM1-2 orientation of this E2P$_{autoinhibited}$ "open" state is different from the active form of ATP8B1 we collected cryo-EM data of the C-terminally truncated ATP8B1 trapped in an E2P conformation with BeF$_x$ in presence of a POPC lipid substrate (Figs. 1 and 3). In the resulting E2P$_{active}$ conformation, the transmembrane bundle displays the same organization as the E2P$_{autoinhibited}$ "open" conformation with the lipid groove being open and filled with PC (Fig. 3 and Supplementary Fig. 19b). Interestingly, in the C-terminally truncated E2P$_{active}$ conformation, no density associated with the N-terminal tail could be observed in the cryo-EM map.

This suggests that the N-terminal tail only interacts with the core protein in the presence of the autoinhibitory C-terminal tail. Given the previously described strong synergistic effect of the N-terminus on C-terminal tail inhibition[26], our structural data suggest that the N-terminal tail stabilizes the C-terminal tail in the bound form, thus reinforcing autoinhibition.

## PI(3,4,5)P$_3$ binding site
As previously shown, C-terminally truncated ATP8B1 exhibited strong stimulation by different phosphoinositides in the presence of a lipid substrate[26] (Fig. 4a). Most of our cryo-EM data have been obtained in presence of PI(4,5)P$_2$, however no clear density for this lipid was observed in the density maps. To identify the phosphoinositide binding site in ATP8B1 we therefore determined the cryo-EM structures of the two full-length E2P$_{autoinhibited}$ states and the E2P$_{active}$ state in the presence of PI(3,4,5)P$_3$, for which ATP8B1 has a higher apparent affinity[26]. For the three structures, a lipid density could be observed in a positively charged cavity formed by TM5, TM7, TM8, and TM10 (Fig. 4b, Supplementary Figs. 21–22a), similar to the PI(4)P and PI(3,5)P$_2$ binding pockets of Drs2p[12,32] and ATP13A2[34], respectively (Supplementary Fig. 21). The observed density is weaker compared to the surrounding protein, potentially indicating partial occupancy of the phosphoinositide binding site and/or flexibility of the ligand.

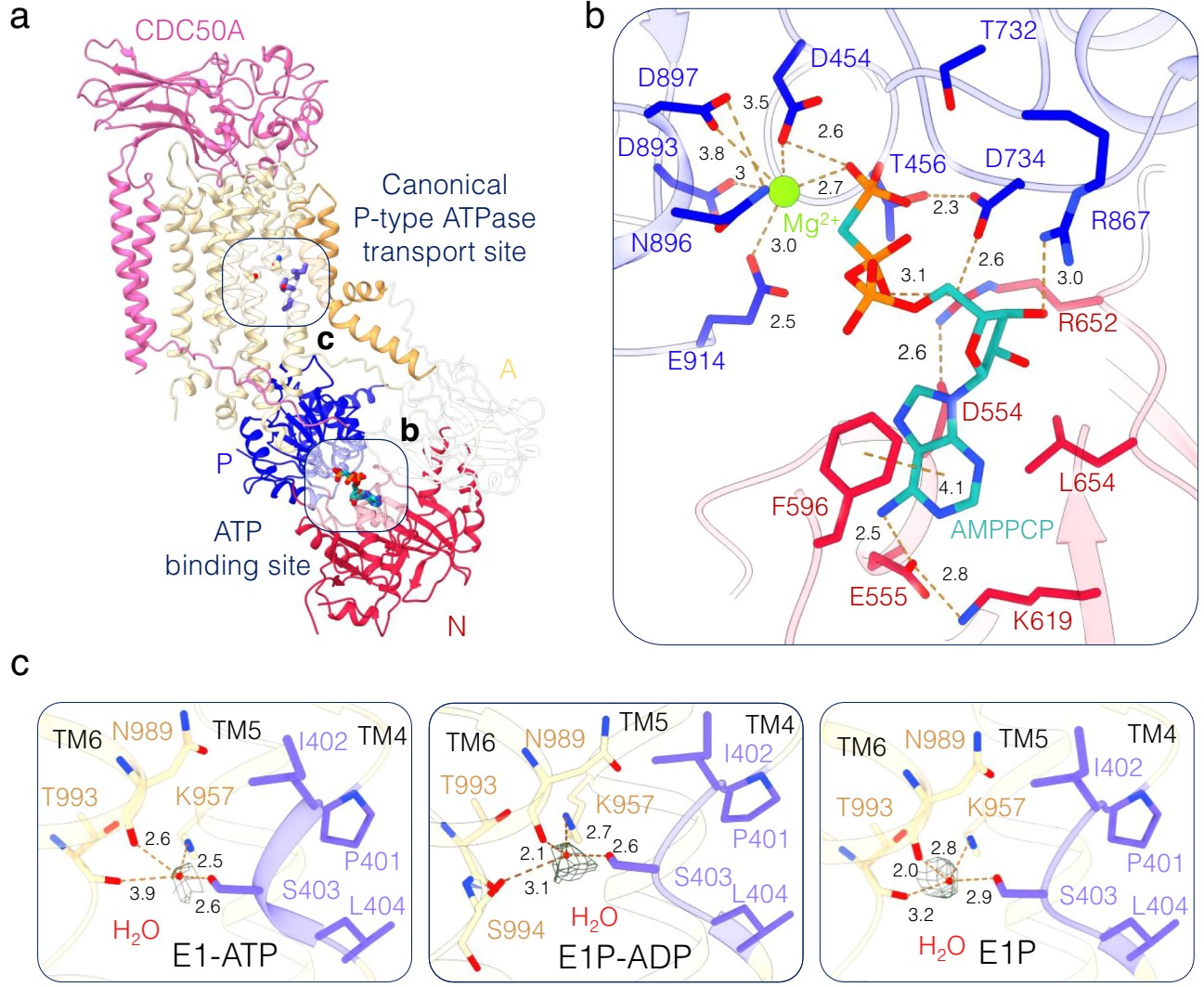

**Fig. 2 | ATP binding and the presence of a water molecule in the core of the lipid transport site. a** ATP8B1-CDC50A structure in a E1-ATP conformation with bound AMPPCP. Color code as Fig. 1. **b** Close-up view of the AMPPCP binding site including a Mg²⁺ ion with coordinating residues as sticks. **c** Detailed views of a part of canonical P-type ATPase transport site of ATP8B1, corresponding to the not yet formed lipid transport site in the E1-ATP (left, contour map: 4.5σ), E1P-ADP (middle, contour map: 9.15σ) and E1P (right, contour map: 7.27σ) conformations, all revealing the presence of a water molecule (carving distance 1.1 Å; for the full EM maps of the region see Supplementary Fig. 17) The water molecule associated EM map is shown in mesh and the P4-ATPase specific PISL motif of TM4 is shown in purple.

Therefore, we performed molecular dynamics (MD) simulations of two systems based on the structure of the E2P$_{autoinhibited}$ "closed" conformation. One with PI(3,4,5)P$_3$ and another with a non-phosphorylated PI molecule, matching the position of the modeled PI(3,4,5)P$_3$ molecule as a starting point. Both systems contained a native-like lipid bilayer composition mimicking the plasma membrane. We then ran a set of five independent 600-ns-long simulations for each system and analyzed the dynamics and interactions between the lipids and the surrounding protein residues. During the equilibration procedure, both lipids slightly shifted from the refined position in the cryo-EM structures (RMSD between carbon atoms of inositol and phosphate atoms: 1.90 Å, Supplementary Fig. 22b). However, both PI(3,4,5)P$_3$ and PI remained bound within the pocket formed by TM5, 7, 8 and 10 during the entire 600 ns in all simulations, suggesting stable binding of the two lipids within the cavity. To analyze the interaction between the protein and the lipids, we calculated hydrogen-bonded populations over the last 200 ns of each trajectory. Although the total number of interacting protein residues is similar for the two systems (12 and 15 for PI(3,4,5)P$_3$ and PI, respectively), we found that PI(3,4,5)P$_3$ forms a more stable contact with the protein compared to PI, as

indicated by an overall increase of hydrogen-bond frequency (Fig. 5a). For PI(3,4,5)P$_3$, the most frequent contacts are established with residues R1032 (0.97), R1164 (0.96) and R952 (0.88). The same residues are also the most frequent interaction partners for PI but with substantially lower populations: R1032 (0.59), R1164 (0.19), and R952 (0.59). This supports a model for the lipid-protein interactions at the PI(3,4,5)P$_3$ binding site based primarily on the interactions between positively charged arginine and lysine residues and the negatively charged phosphates on the lipid headgroup.

Next, we studied the dynamics of PI and PI(3,4,5)P$_3$ in the cavity by monitoring the positions of the center of mass (COM) of the inositol ring relative to the TM region surrounding the lipid-binding site. We found that the hydrogen-bonding patterns are closely related to the dynamics of the headgroups of the two different lipids (Fig. 5b, c): PI shows substantially higher mobility in all three directions, whereas PI(3,4,5)P$_3$ moves mainly along the z-axis (perpendicular to the membrane plane) inside the cavity. To visualize this, we clustered[35] the different binding poses based on the orientations of the inositol. For PI, we found a total of 43 clusters with 19%, 13%, and 13% of the structures populating the top three clusters (45% combined),

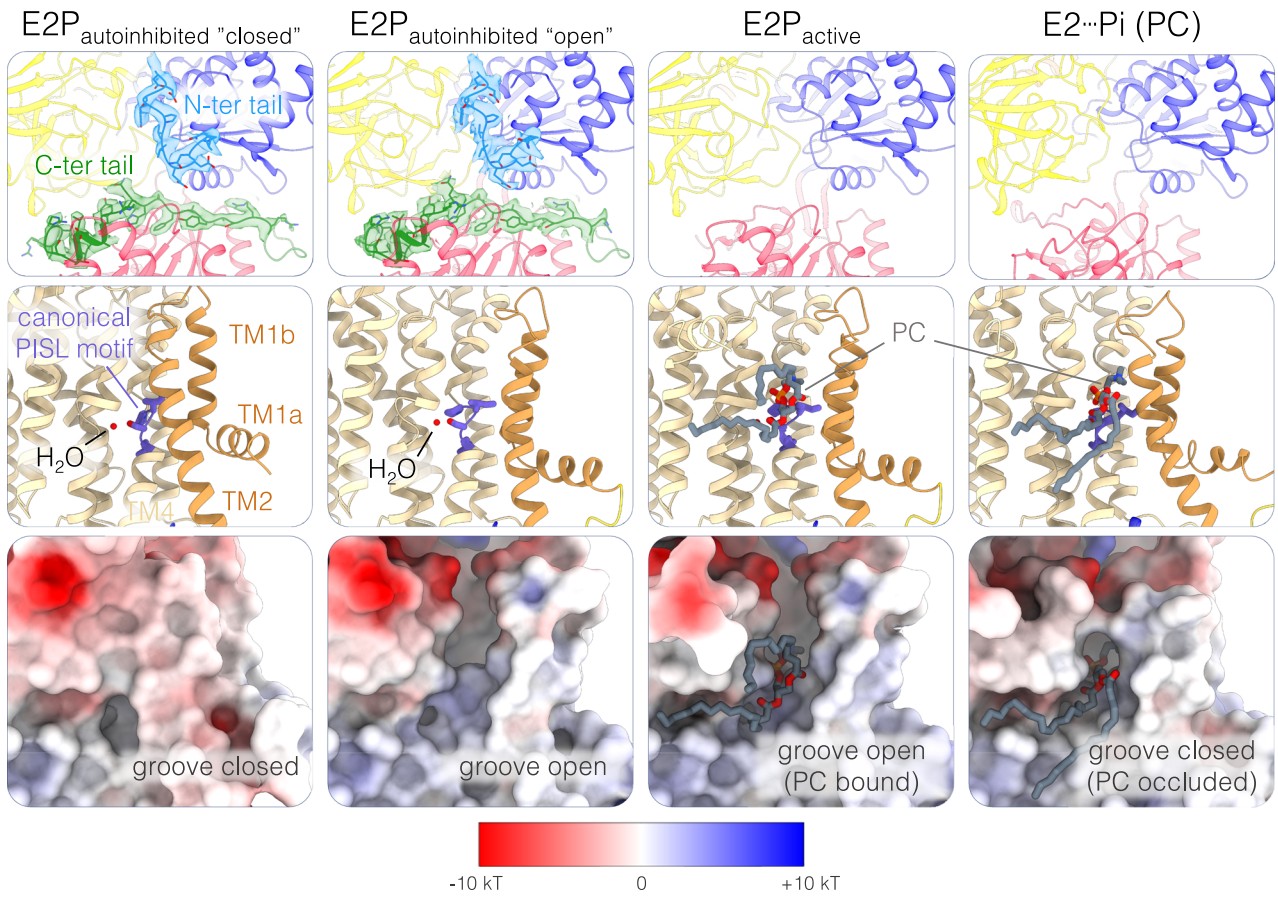

**Fig. 3 | ATP8B1 autoinhibition mechanism and lipid groove opening.** Close-up view of the cytosolic domains with the ATP8B1 N- and C-terminal tails shown with their associated cryo-EM density when present (top row, contour map: 4.17σ and 5.47σ for the E2P_autoinhibited "closed" and "open" conformations, respectively) and the lipid-binding site in cartoon representation (middle row) or as electrostatic surface (bottom row) in the different E2 conformations. Color code as in Fig. 1, with the N-terminal tail of ATP8B1 colored in light blue.

reflecting the large mobility of PI (Fig. 5b–e, Supplementary Fig. 22c). In comparison, we found only a total of 9 clusters for PI(3,4,5)P₃ with 56%, 26%, and 10% of the structures in the top three clusters (92% combined); in line with a more stable binding of PI(3,4,5)P₃. When we examined the solvation properties of the cavity with respect to the bound lipid, we observed on average more water molecules inside the cavity with PI(3,4,5)P₃ bound ($57 \pm 9$) than with PI bound ($47 \pm 7$) (Supplementary Fig. 22d). Additionally, our simulations indicate that on average $2.7 \pm 1.0$ K$^+$ ions populate the binding site when a PI(3,4,5)P₃ is present, whereas no K$^+$ ions are observed in the presence of PI (Supplementary Fig. 22e). This suggests that the PI(3,4,5)P₃ head group binds additional counterions, despite numerous positively charged residues at the binding site. Altogether, our simulations suggest that PI(3,4,5)P₃ binds in a more stable fashion, where dynamics are restricted to mostly "up and down" movements (along a z-axis perpendicular to the membrane plane), whereas PI exhibits increased 3D mobility.

**The transport site of ATP8B1 can accommodate a wide range of glycerophospholipids**

The transport substrate lipid POPC observed in the E2P_active structure adopts a similar conformation as observed for other substrate-bound P4-ATPases[30,36]. The PC headgroup is positioned at the end of a groove formed by TM1, 2, 3, and 4 (Figs. 3 and 6a). The acyl chains of the lipid are not interacting with the protein. Strikingly, the cavity surrounding the choline moiety formed by residues from TM1 and TM2 is wider than in other P4-ATPases characterized so far (Fig. 6a and Supplementary Fig. 23). We therefore hypothesized that with such a cavity,

ATP8B1 should be able to bind a broad range of phospholipid headgroups. To test this hypothesis, we evaluated the ability of the main phospholipids found in eukaryotic membranes to stimulate the ATPase activity of ATP8B1 (Fig. 6b). C-terminally truncated ATP8B1 was incubated with increasing lipid/DDM ratios, and a saturating concentration of PI(3,4,5)P₃. All tested glycerophospholipids stimulated ATP8B1 ATPase activity, whereas SM did not (Fig. 6b). Lipids with smaller head groups (PE, PG, PC, or PA) seemed to be more activating than the negatively charged PS or the bulky PI.

To better understand how ATP8B1 can accommodate such different lipid substrates, we determined the structure of the state associated with lipid occlusion and dephosphorylation using different substrate lipids. For this purpose, we used vanadate as an inhibitor that mimics the transition state of dephosphorylation of the catalytic aspartate. In this E2•••Pi lipid-occluded conformation, the A-domain has rotated and the catalytic E234 residue, from the conserved DGET motif of the A-domain, now coordinates a water molecule for in-line attack of the aspartyl phosphate moiety in the P-domain. The vanadate most likely adopts a VO₃⁻ planar conformation (metavanadate), further ligated by the aspartate and the water molecule, mimicking the phosphate leaving group[37] (Fig. 1). In this conformation, TM1 and 2 now enclose the substrate lipid, blocking the entry groove to prevent the entry of a new lipid (Fig. 3). We collected cryo-EM data sets in the presence of three lipid substrates spanning different chemical properties: PC as a standard lipid, a negatively charged PS, and a bulky PI (Fig. 6c–e, Supplementary Fig. 24 and Supplementary Movie 1). The analysis of the lipid occlusion sites in each case revealed the presence of highly ordered water network interacting with the lipid headgroup

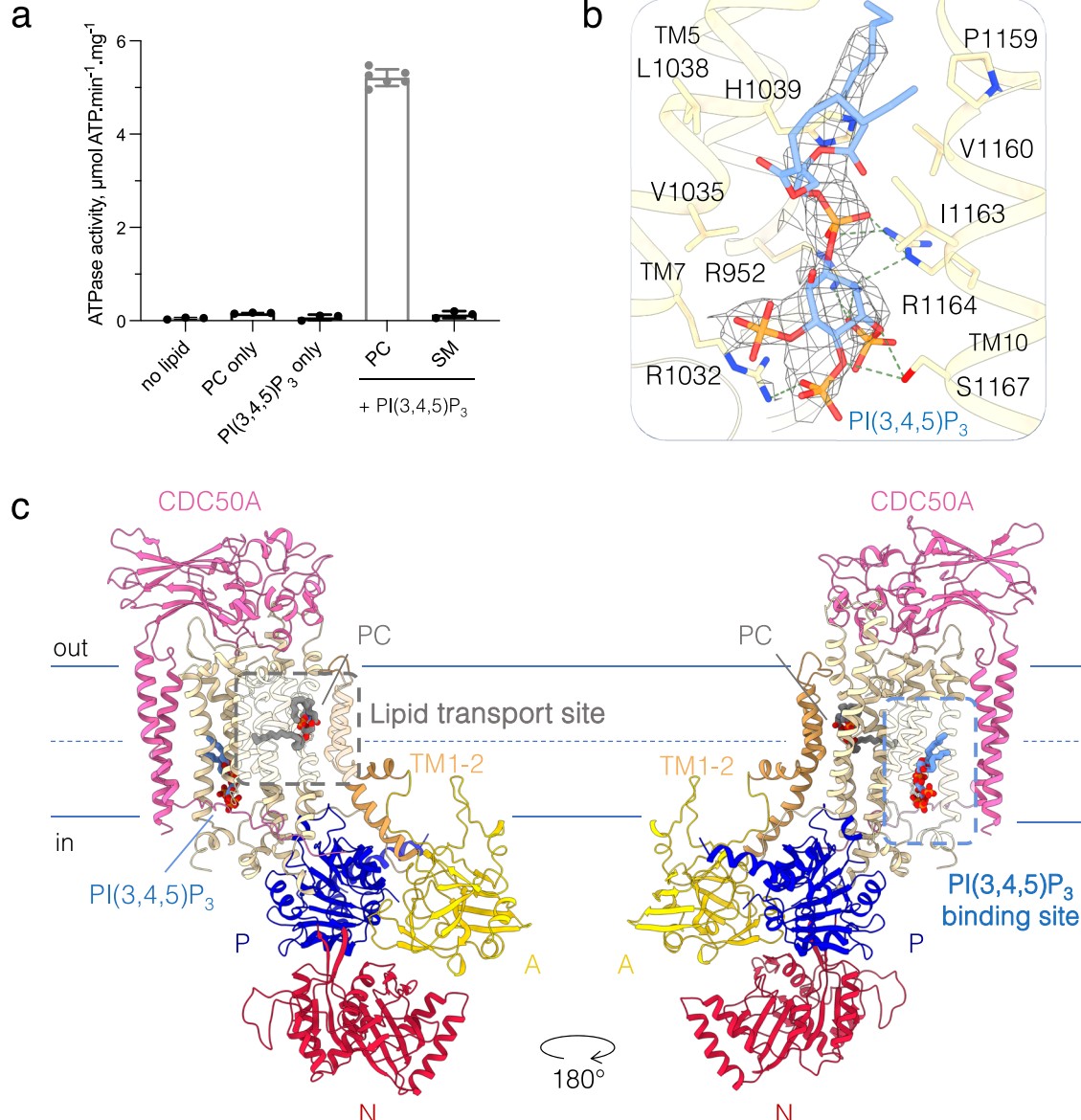

**Fig. 4 | The PI(3,4,5)P$_3$ binding site. a** ATPase activity of the C-terminally truncated ATP8B1-CDC50A complex in the presence of PI(3,4,5)P$_3$ and transport substrate phosphatidylcholine (PC) at 37 °C. The activity was measured in the presence of 1 mg.mL$^{-1}$ DDM (2 mM), 0.17 mg.mL$^{-1}$ PC or sphingomyelin (SM), and, when indicated, with a saturating concentration of 0.025 mg.mL$^{-1}$ PI(3,4,5)P$_3$. The data plotted correspond to the mean ± SD of 3 replicate experiments (6 for PC + PI(3,4,5)P$_3$) from the same purification. **b** Close-up view of the PI(3,4,5)P$_3$ binding site and its associated EM density (E2P$_{autoinhibited}$ "closed", 2.8σ). **c** Overall structure of ATP8B1-CDC50A complex in the E2P$_{active}$ conformation with PC and PI(3,4,5)P$_3$ bound. Color code as in Fig. 1.

and the protein (Fig. 6f). Interestingly, the structures showed that all three lipids are occluded in a similar manner, and side chains of the residues forming the occlusion site adopted virtually identical structures despite the presence of the charged carboxylic acid moiety of PS or the bulkier inositol moiety of PI (Fig. 6g). However, the water network differs among the different lipid occlusion sites (Fig. 6h), which could also explain the differences observed when comparing the ATPase activities. One may speculate that the organization of this water network is part of the triggering event of the allosteric transmission through TM2 that enables the A-domain rotation required for the dephosphorylation event.

**Sn-2 ester moiety is critical for substrate recognition**

Given that all major glycerophospholipids tested were able to stimulate ATPase activity, unlike sphingomyelin, and can be recognized and occluded by the ATP8B1-CDC50A complex, we then sought to identify which part of the lipid is essential for substrate recognition. In the outward-open E2P$_{active}$ state, the phosphoglycerol backbone of PC was found close to the PISL motif of ATP8B1. Interestingly, in this conformation, a water molecule is observed interacting with the S403 side chain (Fig. 7a, Supplementary Fig. 24a). For the E2··Pi lipid-occluded state, however, the lipid is bound deeper in the pocket and the water molecule now displaced (Fig. 7b, Supplementary Fig. 24b–d). In the E2··Pi lipid-occluded conformation, the S403 side chain is less well defined but the different possible rotamers of the S403 can now interact with the phosphate or with the sn-2 ester carbonyl more tightly (Fig. 7b, Supplementary Fig. 24b–d). To evaluate the importance of the sn-2 ester carbonyl interaction, we tested the ability of lipid analogs to stimulate the ATPase activity of the ATP8B1-CDC50A complex (Fig. 7c). All lipids tested share the same choline headgroup and only differ in their chemical linkage between the phosphate group and the acyl chains. Plasmanylcholine, which contains an ether bond in sn-1 and an ester bond in sn-2, appeared to stimulate the ATPase activity almost as efficiently as PC (ester-PC). In contrast, ether PC,

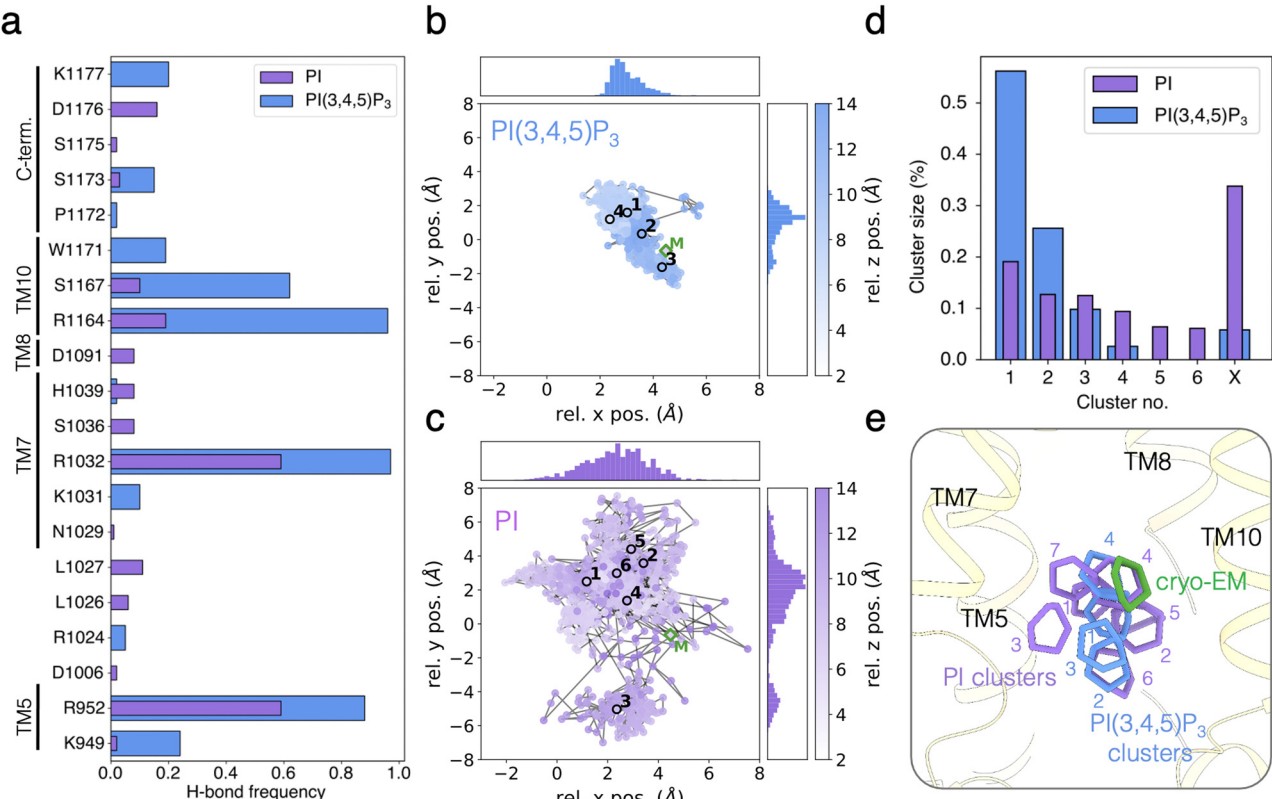

**Fig. 5 | Molecular dynamics studies of the PI(3,4,5)P₃ binding site. a** Hydrogen bonding (H-bond) frequencies between protein and the PI(3,4,5)P₃ (blue) and PI (purple) headgroup from the last 200 ns of five MD trajectories of each system. **b, c** Relative x, y, and z positions of the center of mass (COM) of the inositol ring to the COM of selected Cα atoms of cavity-lining protein residues used as reference (see **e**) showing different mobility of the PI(3,4,5)P₃ (blue; **b**) and PI (purple; **c**) headgroups. The highlighted points and numbers correspond to the cluster centroids in **e** and M indicating the initial position of the PI(3,4,5)P₃ molecule as modeled and refined in the cryo-EM map of the E2P_autoinhibited "closed" conformation. **d** Clustering of inositol ring coordinates from the last 200 ns of each trajectory for PI(3,4,5)P₃ (blue) and PI (purple). **e** Structural visualization of the cluster centroids from clustering PI(3,4,5)P₃ (blue) and PI (purple) inositol poses in the binding cavity (wheat). The inositol ring position as refined in the cryo-EM map of the E2P_autoinhibited "closed" with PI(3,4,5)P₃ conformation is shown in green. Only centroids from clusters containing ≥5% of the clustered structures are shown.

which lacks the ester bonds in sn-1 and sn-2 positions, failed to efficiently activate ATP8B1, achieving only 10% of the activity observed in the presence of PC at an equivalent lipid/DDM ratio. As previously described[26], lyso-PC, which lacks the fatty acid chain in sn-2 position, can stimulate the ATPase activity, but to a much lower extent than PC. Together, our data strongly suggest that the presence of an acyl chain in sn-2 position, and more specifically of an ester bond in the sn-2 position, is a critical determinant for lipid recognition by ATP8B1.

## Discussion

### A water molecule at the transport site in P4- and P5-ATPases

Biochemical and structural studies of P4-ATPases have shown substrate-independent phosphorylation from ATP unlike for cation pumps where phosphorylation is tightly coupled to cation binding and occlusion. Thorough characterization of bovine ATP8A2-CDC50A and yeast Drs2p-Cdc50p also showed that phosphorylation was not dependent on a range of cations and quite insensitive to pH[11,38]. Lipid transport of ATP8A2 on solid-supported membranes revealed a net outward charge transfer upon PS transport, also supporting substrate-independent phosphorylation[39,40]. Our studies agree with this model, but we additionally identify a neutral water molecule in the transport site of P4-ATPases, which interacts with polar side chains in the absence of a substrate lipid headgroup (Fig. 2c, Supplementary Fig. 17). Interestingly, the water molecule stays bound in the outward-open E2P_active form of ATP8B1, where an incoming substrate lipid makes first interactions with the lipid-binding site. The water molecule is then chased upon occlusion of the substrate lipid, and the lipid headgroup

directly interacts with the S403 side chain of the PISL motif, i.e. replacing the water molecule. The role and dynamics of the water structure and how it is involved in the P4-ATPase cycle invite further studies by e.g. MD simulations. Interestingly, the structural and functional data obtained on P5A- and P5B-ATPases such as human ATP13A2 also show that substrate transport (of transmembrane alpha helices and polyamines, respectively) is coupled to dephosphorylation, whereas phosphorylation is independent of other substrates, i.e., similar to P4-ATPases[31,34,41–43]. Congruent with this, the polyamine transport site of human ATP13A2 also binds water molecules in the E1 to E1P half-cycle[31] (Supplementary Fig. 18). From an evolutionary point of view, the presence of a water binding site in P4- and P5-ATPases might result from the modification of a cation site of their last common ancestor, thus acquiring the ability to undergo phosphorylation from ATP in a *de facto* spontaneous mechanism.

### Autoinhibition is compatible with lipid-binding, but not occlusion

Regulation through N- and C-terminal tails is a feature shared among various P-type ATPases, but structurally it was first described for the Drs2p-Cdc50p flippase complex, which revealed how the C-terminal tail occupies the nucleotide-binding pocket and locks the cytoplasmic domains[32]. A similar autoregulation mechanism applies to ATP8A1 and ATP8B1[7,26,27]. C-terminal autoregulation locks the A-domain thereby restricting TM1-2 movements required for the opening of the outward-oriented lipid entry pathway (Supplementary Fig. 20). We show here that despite the presence of the N- and C-terminal tails at the interface

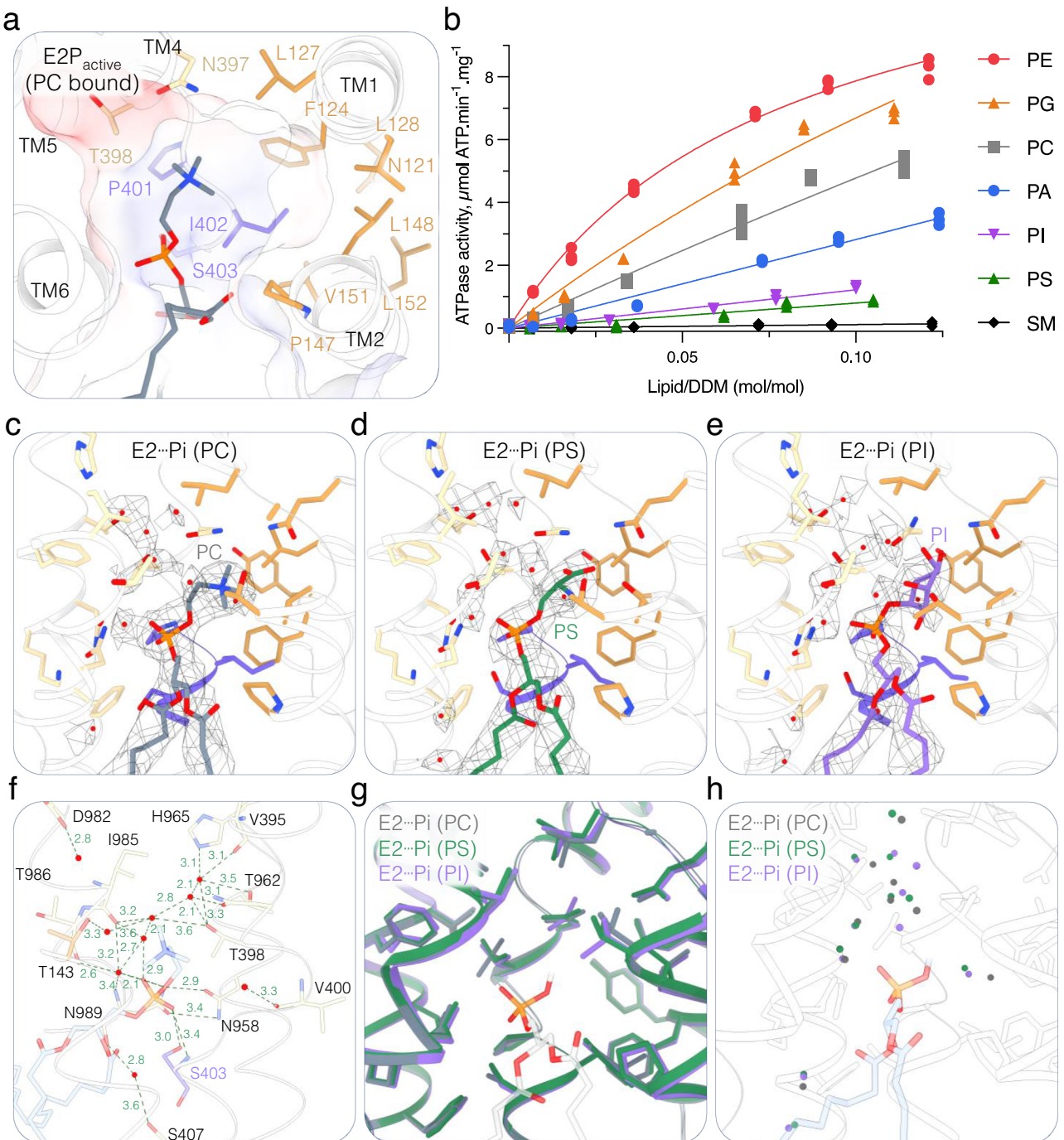

**Fig. 6 | ATP8B1 lipid substrate specificity, binding, and occlusion. a** Close-up view of the lipid-binding site of ATP8B1 in the E2P$_{active}$ state occupied by a phosphatidylcholine (PC) molecule. The cavity formed by the protein around the lipid headgroup is shown as electrostatic surface. **b** ATPase activity of the C-terminally truncated ATP8B1-CDC50A complex in the presence of phosphatidylethanolamine (PE), phosphatidylglycerol (PG), phosphatidylcholine (PC), phosphatidic acid (PA), phosphatidylinositol (PI), phosphatidylserine (PS) or sphingomyelin (SM). The activity is measured at 37 °C with different lipid concentrations but in the presence of an invariant concentration of 1 mg.mL$^{-1}$ DDM (2 mM) and 0.025 mg.mL$^{-1}$ PI(3,4,5) P$_3$. The data plotted correspond to 3 replicate experiments from the same purification. Data were fitted to a hyperbolic function. **c–e** Close-up view of the lipid occlusion site of ATP8B1 in the E2···Pi lipid-occluded states occupied by PC (**c**), PS (**d**), or PI (**e**). The lipid-associated EM density is shown as mesh (5σ (**c**) 4.11σ (**d**), 5.04σ (**e**)). **f** Detailed view of the water molecule network observed in the lipid occlusion site of the E2···Pi (PC) structure of ATP8B1-CDC50A complex. **g** Cartoon representation of the structural alignment of the lipid occlusion site observed in the different E2···Pi lipid-occluded structures obtained with PC (gray), PS (green), or PI (purple). The PS molecule is shown without its headgroup to underline the lipid position in the three E2···Pi structures. **h** Comparison of the position of the water molecules observed in the lipid occlusion site.

of the cytoplasmic domains, the A-domain maintains some level of movement and TM1-2 can rotate to open the lipid entry pathway. However, lipid occlusion and dephosphorylation cannot occur. Thus, our data indicate that the E2P$_{autoinhibited}$ state of ATP8B1 explores both open and closed conformations, but only the open state can initiate

lipid binding. In our experimental conditions (4 °C in LMNG/CHS/ PI(3,4,5)P$_3$ mixed micelles), the closed conformation seems to be favored, but the equilibrium could be poised toward the open conformation in a native membrane. Further structural investigations of ATP8B1 in a more native environment will be needed to evaluate

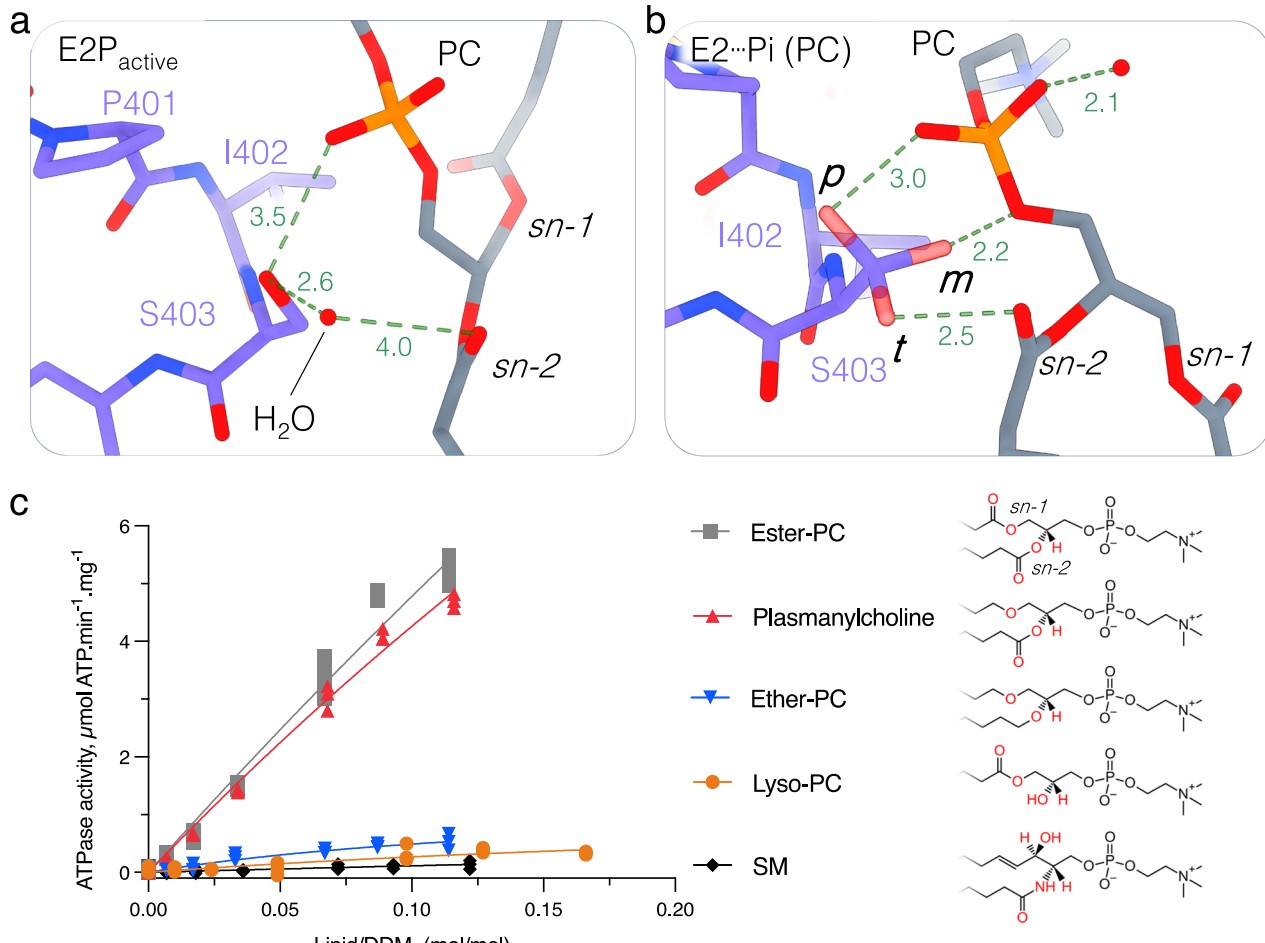

**Fig. 7 | The sn-2 ester bond of the transport lipid is essential for substrate recognition. a, b** Close-up view of lipid and water interactions with the S403 of the PISL motif of TM4 in the E2P$_{active}$ (**a**) and E2···Pi (PC) (**b**) conformations. The different rotamers of S403 from Coot's rotamer library are designated as p, m, and t (see Supplementary Fig. 24 for associated cryo-EM maps). **c** ATPase activity of the C-terminally truncated ATP8B1-CDC50A complex in the presence of different phospholipids with a choline headgroup (phosphatidylcholine-based lipids (PC) or sphingomyelin (SM)). The activity is measured at 37 °C with different lipid concentrations but in the presence of an invariant concentration of 1 mg.mL$^{-1}$ DDM (2 mM) and 0.025 mg.mL$^{-1}$ PI(3,4,5)P$_3$. The data plotted correspond to 3 replicate experiments from the same purification.

## ATP8B1 activation by phosphoinositides

ATP8B1 is directly activated by PI(3,4,5)P$_3$, although other phosphoinositides such as the more abundant PI(4,5)P$_2$ can also promote its ATPase activity[26]. From the cryo-EM data we identified a PI(3,4,5)P$_3$ binding site in a positively charged cavity lined by TM8, 9, and 10. Additionally, the binding site geometry remains unchanged through all conformations of the catalytic cycle, suggesting that the regulatory lipid binds in a conformation-independent manner (Supplementary Fig. 25a). Our MD data show a clear interaction and stabilization of PI(3,4,5)P$_3$ compared to PI in the identified pocket. However, the cryo-EM density map corresponding to the PI(3,4,5)P$_3$ molecule was weaker compared to the rest of the protein which suggests flexibility or a partial occupancy of the site. This might also reflect the difference in experimental conditions between ATPase activity assays (37 °C in DDM) and cryo-EM sample preparation (4 °C in LMNG). Although not addressed here, the physiological role of this activation remains unclear. Given the localization of ATP8B1 at the apical membrane of epithelial cells, PI(4,5)P$_2$ represents the most abundant phosphoinositide compared to PI(3,4,5)P$_3$[44], which on the other hand displays a higher affinity[26]. Synthesis and degradation of phosphoinositides is a

highly dynamic process related to cell signaling, and the cascades regulating ATP8B1 have not been identified yet. Moreover, questions remain open on the exact mechanism by which phosphoinositides activate ATP8B1, and which specific steps of the catalytic cycle are stimulated. For Drs2p-Cdc50p, PI(4)P stabilizes an amphipathic helix, destabilizes binding of the autoinhibitory C-terminal tail, and stimulates dephosphorylation of E2P coupled to lipid occlusion and transport[32,45]. Yet, C-terminally truncated Drs2p-Cdc50p requires PI(4)P for full activity[32,46]. Similarly, truncated versions of ATP8B1 also require phosphoinositides for activation indicating that if phosphoinositides play a role in relief of autoinhibition, they also have an additional function. Our studies do not provide immediate clues on the direct link between phosphoinositide binding and activation of ATP8B1, but it is likely linked to biased dynamics favoring transport substrate recognition and/or occlusion in the E2 half-cycle, as we could previously observe that the ATP-dependent E1 to E2P half-cycle is not modulated by phosphoinositides[26]. In our study we could observe that PI(3,4,5)P$_3$ interacts tightly with the R952, which is located five residues after K957 on TM5, the latter being close to the lipid transport site and in interaction with the previously described water molecule (Supplementary Fig. 25b). However, comparison of our MD simulations did not reveal striking differences in R952 and TM5 orientation, whether PI or PI(3,4,5)P$_3$ is bound. Future work should aim to elucidate the

mechanism by which phosphoinositides activate flippases, and to identify the signaling pathways involved in the activation of ATP8B1 in the cell.

## ATP8B1 has a broad lipid specificity

So far, no consensus has been established regarding the transport substrate of ATP8B1[20–27]. Our structures of the E2P$_{active}$ and E2···Pi states bound to lipids show that different lipid head groups may be accommodated at the lipid-binding site and more importantly, that they can be occluded. This is consistent with all of them being able to stimulate ATPase activity (Fig. 6b), and strongly suggests that they are also transport substrates. The different sizes and chemical properties between the lipid headgroups bound at the transport site may lead to differences in kinetics of binding and occlusion and thus explain why they stimulate ATPase activity in a different manner. In comparison, the lipid headgroup binding cavity of other P4-ATPases is more restrictive compared to ATP8B1 (Supplementary Fig. 23). A comparison of ATP11C and ATP8B1 in lipid-bound E2P states (i.e., where lipid recognition takes place prior to occlusion) shows that TM1 and TM2 are oriented differently. In ATP11C, TM1-2 are closer to TM3-4, thus diminishing the size of the groove for lipid-binding (Supplementary Fig. 26 and Supplementary Fig. 27). In line with this, a screen for glycerophospholipids showed that for ATP11C, only PS and PE stimulated ATPase activity and were transported[24,36,47]. A sequence alignment of ATP8B1 to PS/PE-specific flippases such as ATP11C shows a high degree of conservation at the lipid-binding cavity (Supplementary Fig. 27). Only few residues differ significantly, in particular W390, but surprisingly, they are not located directly at the lipid headgroup binding site. Rather, the bulky W390 side chain keeps TM1-2 at a further distance from TM3-4 and would provide a wider and more permissive groove for lipid headgroup recognition and occlusion. Previous studies on ATP8A1 showed that ether-PS is not transported across the erythrocyte membrane[48]. Here, we provide evidence that the carbonyl group of the sn-2 ester bond plays a critical role in substrate-induced ATPase activity in ATP8B1, most likely due to the interactions it forms with S403 of the conserved PISL motif (Fig. 7b). Noteworthy, a S403Y mutation has been found in patients suffering from PFIC1[29], and this mutation has been shown to dramatically diminish the ATPase activity of the purified ATP8B1-CDC50B complex[27]. Sphingosine-based lipids, like SM, are not substrates of ATP8B1. In sphingolipids, the sn-2 ester bond is replaced by an amide bond, which engages in intra-molecular hydrogen bonds with the phosphate group[49]. In our three different E2···Pi lipid-occluded structures, the phosphate moiety of the lipid headgroup interacts simultaneously with S403 and a highly ordered water molecule (Fig. 7b and Supplementary Movie 1). Hence, the internal H-bond of SM would disfavor the interactions observed for glycerophospholipids. Thus, our study broadens the lipid specificity of ATP8B1 to sn-1-mono-ether phospholipids, in line with a recently identified link between ATP8B2 and plasmalogens homeostasis[50]. It should be mentioned that we cannot exclude the possibility that sn-2-mono-ether phospholipids could also be recognized by ATP8B1, although probably not as effectively as sn-1-mono-ether phospholipids due to the proximity of the sn-2 ester bond with S403 observed in our structures. Transport studies of such non-natural sn-2-mono-ether phospholipids by P4-ATPases could potentially pave the way to the development of lipid-based drugs that would be better recognized compared to previously evaluated anti-cancer alkylphosphocholine analogs such as miltefosine and edelfosine[21].

In summary, we present a series of nine cryo-EM structures of the ATP8B1-CDC50A complex with accompanying functional studies and MD simulations, deciphering most of its catalytic cycle. The structures highlight a so far unrecognized water molecule bound at the lipid transport site during the autophosphorylation part of the catalytic cycle. Additionally, we observe two different autoinhibited states, a closed and an outward-open conformation which are likely in a dynamic equilibrium. Finally, our work reveals important determinants of lipid specificity in P4-ATPases and explains earlier discrepancies about the ATP8B1 transport substrate. Future work should aim at deciphering the subtle but crucial effects of phosphoinositide binding and activation.

## Methods

### Co-expression of the hATP8B1-hCDC50A complex

hATP8B1 (Uniprot: O43520; A1152T natural variant) and hCDC50A (Uniprot: Q9NV96) were co-expressed in a protease deficient *Saccharomyces cerevisiae* W303.1b/*Δpep4* (*MATα, leu2-3, his3-11, ura3-1, ade2-1, Δpep4, can^r, cir^+*) yeast strain as previously described[26]. To generate a C-terminal cleavable construct, an overlap extension PCR strategy was used to insert 8 aminoacids forming a HRV 3C protease site (LEVLFQGP, see Supplementary Table 2 for primer sequences) in between R1184 and K1186 of ATP8B1 sequence. Yeast strains and plasmids are available upon request.

### Purification of the ATP8B1-CDC50A complex

Yeast cells were resuspended in buffer A (50 mM Tris-HCl pH 7.5, 1 mM EDTA, 0.6 M sorbitol) supplemented with protease inhibitors (2 mM PMSF, 2 µg.mL$^{-1}$ leupeptine, 2 µg.mL$^{-1}$ pepstatin, 2 µg.mL$^{-1}$ chymostatin). The cells were subsequently broken with 0.5 mm glass beads using bead beaters. The crude extract was then spun down at 3,000 g for 30 min at 6 °C, to remove cell debris and nuclei. The membrane fraction was pelleted at 100,000 g for 90 min at 4 °C. The resulting yeast membrane pellets were finally resuspended at about 50 mg.mL$^{-1}$ of total protein in buffer A supplemented with protease inhibitors (1 mM PMSF, 1 µg.mL$^{-1}$ leupeptine, 1 µg.mL$^{-1}$ pepstatin, 1 µg.mL$^{-1}$ chymostatin). Membranes were diluted to 5.5 mg.mL$^{-1}$ of total protein in ice-cold buffer B (50 mM MOPS-Tris at pH 7, 100 mM KCl, 1 mM DTT, 20% (w/v) glycerol and 5 mM MgCl$_2$), supplemented with protease inhibitors (1 mM PMSF, 1 µg.mL$^{-1}$ leupeptine, 1 µg.mL$^{-1}$ pepstatin, 1 µg.mL$^{-1}$ chymostatin). The suspension was then supplemented with 15 mg.mL$^{-1}$ DDM and 5 mg.mL$^{-1}$ CHS and incubated 1 h at 4 °C. Insoluble material was pelleted by centrifugation at 125,000 g for 1 h at 4 °C. The supernatant, containing solubilized proteins, was applied onto a streptavidin-sepharose resin and incubated for 2 h at 6 °C. The resin was washed twice with six resin volumes of ice-cold buffer B supplemented with 0.2 mg.mL$^{-1}$ LMNG and 0.02 mg.mL$^{-1}$ CHS and protease inhibitors (1 mM PMSF, 1 µg.mL$^{-1}$ leupeptine, 1 µg.mL$^{-1}$ pepstatin, 1 µg.mL$^{-1}$ chymostatin). The resin was then washed thrice with six resin volumes of ice-cold buffer B supplemented with 0.1 mg.mL$^{-1}$ LMNG and 0.01 mg.mL$^{-1}$ CHS. To delipidate the purified ATP8B1-CDC50A complex, each washing step was performed in batch with a prolonged incubation time of at least 15 min, to allow the slowly diffusing lipids to be incorporated in detergent micelles[51]. Elution was performed by the addition of 60 µg of purified TEV protease per mL of resin and overnight incubation at 6 °C. The eluted fraction was concentrated using a Vivaspin unit (100 kDa MWCO) prior to injection on a size-exclusion Superose 6 10/300GL increase column equilibrated with buffer C (50 mM MOPS-Tris pH 7, 100 mM KCl, 1 mM DTT, 5 mM MgCl$_2$) supplemented with 0.03 mg.mL$^{-1}$ LMNG and 0.003 mg.mL$^{-1}$ CHS. For 3C protease site-containing version of ATP8B1, 100 µg of purified 3C protease (1:20 mol:mol ATP8B1:3C protease) was added to the concentrated sample and incubated 18 h at 6 °C and 1 h at 30 °C prior to size-exclusion chromatography.

### Lipids

Lipids used for ATPase activity measurement and cryo-EM sample preparation were POPC (Avanti Polar Lipid Inc., 850457 P), POPS (Avanti Polar Lipid Inc., 840034 P), soy PI (Avanti Polar Lipid Inc., 840044 P), POPA (Avanti Polar Lipid Inc., 840857 P), POPE (Avanti Polar Lipid Inc., 850757 P), Egg PG (Avanti Polar Lipid Inc., 841138 P), Egg Sphingomyelin (Avanti Polar Lipid Inc., 860061 P), 18:0 Lyso-PC

 

(Avanti Polar Lipid Inc., 855775 P), C16-18:1 Sn-1 Ether PC ( = Plasmanylcholine, Avanti Polar Lipid Inc., 878112 P), C16:0-18:1 Diether PC (Avanti Polar Lipid Inc., 999983 P), Brain PI(4,5)$P_2$ (Avanti Polar Lipid Inc., 840046 P) and diC16 PI(3,4,5)$P_3$ (Echelon, P-3916).

## Cryo-EM sample preparation

Different conformations of the ATP8B1-CDC50A complex were stabilized in the following way: For the determination of the E1-ATP (AMPPCP), E1P-ADP (ADP+AlF$_4^-$), E1P (AlF$_4^-$), E2···Pi (PC) (VO$_3^-$ + POPC), E2···Pi (PS) (VO$_3^-$ + POPS) and E2···Pi (PI) (VO$_3^-$ + soy PI) states, the C-terminally truncated construct was incubated with PI(4,5)$P_2$ 0.05 mg.mL$^{-1}$ during 3C protease treatment. The sample was then injected on a size-exclusion Superose 6 10/300GL increase column equilibrated with buffer C supplemented with 0.03 mg.mL$^{-1}$ LMNG and 0.003 mg.mL$^{-1}$ CHS. The purified complex was then concentrated to 0.8-5 mg.mL$^{-1}$ and incubated with PI(4,5)$P_2$ 0.0015 mg.mL$^{-1}$ and respectively with 1 mM AMPPCP / 3 mM ADP + 2 mM AlCl$_3$ + 10 mM NaF / 2 mM AlCl$_3$ + 10 mM NaF / 1 mM Na$_3$VO$_4$ + 0.003 mg.mL$^{-1}$ POPC / 1 mM Na$_3$VO$_4$ + 0.003 mg.mL$^{-1}$ POPS / 1 mM Na$_3$VO$_4$ + 0.003 mg.mL$^{-1}$ soy PI for 1 h on ice. For the E2P$_{active}$ state: the C-terminally truncated construct was incubated with 0.01 mg.mL$^{-1}$ POPC and 0.002 mg.mL$^{-1}$ PI(3,4,5)$P_3$ during elution from the streptavidin resin by TEV protease. The sample was then injected on a size-exclusion Superose 6 10/300GL increase column equilibrated with buffer C supplemented with 0.003 mg.mL$^{-1}$ POPC and 0.0015 mg.mL$^{-1}$ PI(3,4,5)$P_3$. The purified complex was then concentrated to 5 mg.mL$^{-1}$ and incubated with 1 mM BeSO$_4$ and 4 mM KF for 1 h on ice. For the E2P$_{autoinhibited}$ states: the WT full-length construct was incubated with 0.002 mg.mL$^{-1}$ PI(3,4,5)$P_3$ during elution from the streptavidin resin by TEV. After concentration the sample was incubated with 2 mM ATP. The sample was then injected on a size-exclusion Superose 6 10/300GL increase column equilibrated with buffer C supplemented with 0.0015 mg.mL$^{-1}$ PI(3,4,5)$P_3$. For this sample, CHS was omitted from all steps of the purification as we initially hypothesized that CHS might compete with phosphoinositides. The purified complex was then concentrated to 3 mg.mL$^{-1}$. In all cases, 3 μL of the final sample was added to freshly glow-discharged (45 s at 15 mA) C-flat Holey Carbon grids, CF-1.2/1.3–4 C (Protochips), which were subsequently vitrified at 4 °C and 100% humidity on a Vitrobot IV (Thermo Fisher Scientific).

## Cryo-EM data collection

All data were collected on a Titan Krios G3i (EMBION Danish National cryo-EM Facility – Aarhus node) with X-FEG operated at 300 kV and equipped with a Gatan K3 camera and a Bioquantum energy filter using a slit width of 20 eV. Movies were collected using aberration-free image shift data collection (AFIS) in EPU (Thermo Fisher Scientific) as 1.5 second exposures in super-resolution mode at a calibrated physical pixel size of 0.647 Å/pixel (magnification of 130,000x) with a total electron dose of 60 e$^-$/Å$^2$.

## Cryo-EM data processing

For each data set a detailed procedure of the data processing workflow is available in Supplementary Figs. 3–11. Briefly, processing was performed in cryoSPARC[52] (v3). Patch Motion Correction and Patch CTF were performed before low-quality micrographs (e.g. micrographs with crystalline ice, high motion) were discarded. Particles were initially picked using template picking on all movies. Particles were extracted in a 384 or 416-pixel box and Fourier cropped four times to a 96- or 104-pixel box (2.59 Å/pixel). Ab initio references were produced using a subset of all particles selected from 2D class. One protein-like reference and multiple junk references were used in multiple rounds of heterogeneous refinement. Selected particles were then re-extracted in an appropriate pixel box depending on the maximum resolution achieved with non-uniform refinement with a first re-extraction in a 208, 240 or 256-pixel box. The final particle stack was used to perform

a final non-uniform refinement combined with per-particle CTF refinement[53]. The resulting map was used to generate a mask to run 3D variability analysis to investigate sample flexibility and heterogeneity[54]. For the two E2P autoinhibited states, conformational heterogeneity was detected, and the two final particles stack were subjected to a final non-uniform refinement combined with per-particle CTF refinement.

## Model building

The ATP8B1-CDC50A models were built using the previously published structure of the complex (PDB: 7PY4) as templates. The cryo-EM maps were sharpened using the Autosharpen tool in PHENIX[55]. The models were manually generated and relevant ligands were added with COOT[56] before real-space refinement in PHENIX[57]. For real space refinement, hydrogen addition and ligand restraints were generated by the ReadySet tool in PHENIX. Model validation was performed using MolProbity[58] in PHENIX, and relevant metrics are listed in Supplementary Table 1.

## Molecular dynamic simulation and analysis

The starting structures of the ATP8B1/CDC50A complex for the MD simulations were based on the cryo-EM model of the E2P$_{autoinhibited}$ "closed" state (PDB:8OX7). We modeled missing loops (28–62, 72–105, 714–786, and 814–832) using MODELLER[59] (v10.1). The template for loop 2 (72–105) was based on the corresponding AlphaFold2 model[60] because of the presence of a local hairpin structure with reasonable confidence and confirmed by the density observed in the E2P-active conformation. We modeled the phosphorylated D454 with two negative charges and parameterized it as previously described[61]. We kept the Mg$^{2+}$ ion and three disulfide bonds in CDC50A (C91-C104, C94-C102, and C157-C171) but removed glycans. E230 and E843 were protonated and K957 was deprotonated based on predictions using PROPKA3[62] to match the experimental conditions at pH 7. The IP9 molecule in the cryo-EM structure was replaced by a PI or PI(3,4,5)$P_3$, modeled as a SAPI (18:0/20:4) and SAPI34 (18:0/20:4), respectively. We set up the system using the CHARMM-GUI Membrane Builder web tool[63] and parameterized the system with the Charmm36m force field[64] in conjunction with the TIP3 water model[65] using the CHARMM-GUI Input Generator[66]. The protein complex was embedded in a bilayer, and we chose the lipid composition of the cytosolic (inner/lower) leaflet and luminal (outer/upper) leaflet to mimic a mammalian plasma membrane[67,68]. The lower leaflet consists of 15% cholesterol (CHOL), 20% phosphatidylcholine (13% PLPC and 7% POPC), 33% phosphatidylethanolamine (YOPE), 23% phosphatidic acid (YOPA), 3% phosphatidylserine (POPS), and 6.0% phosphatidylinositol (3% SAPI, 1% SAPI14, 1% SAPI2D, and 1% SAPI34). The upper leaflet is composed of 39% cholesterol (CHOL), 46% phosphatidylcholine (31% PLPC and 15% POPC), and 15% sphingomyelin (SM). The membrane spans an area of 14 × 14 nm, totaling about 300 lipids in each leaflet. We solvated the systems with about 92,000 water molecules, neutralized the systems, and matched experimental ion concentrations of 100 mM KCl and 5 mM MgCl$_2$ with corresponding ions. The total atom number was around 370,000 and the equilibrated box dimensions in the x, y, and z dimension were 13.5 nm, 13.5 nm, and 19.9 nm, respectively.

We minimized the systems with 5000 steps of steepest descent and positional restraints on protein backbone side chain atoms and lipid atoms. We performed two steps of equilibration in the NVT ensemble (250 ps each) using the Berendsen thermostat[69] and a constant temperature of 310 K. This was followed by 4 more steps of equilibration in the NPT ensemble of 500 ps each by introducing semi-isotropic pressure coupling using the Berendsen barostat. Positional restraints were gradually released during equilibration steps. We performed the production run in the NPT ensemble at 310 K using the Noose-Hoover thermostat and a coupling constant of 1 ps. The pressure was kept constant at 1 bar with semi-isotropic pressure coupling using the Parrinello-Rahman barostat[70], a 5 ps time coupling constant, and an

isothermal compressibility of $4.5.10^{-5} bar^{-1}$. We applied a 1.2 nm cutoff for the van der Waals interactions using a switching function from 1.0 nm and a cutoff of 1.2 nm for short-range electrostatic interactions while long-range electrostatics were treated by Particle-mesh Ewald summation[71]. The Verlet algorithm was used to generate a pair list with buffer and the time step for integration was 2 fs. For each system (PI bound and $PI(3,4,5)P_3$-bound), we performed a set of five 600 ns long production runs. Finally, we analyzed the last 200 ns of each simulation, pooled the five replicates, and used a frame every 1 ns. All simulations were run using GROMACS[72] (v2020/3). Analysis was performed using GROMACS v 2019/4 and python using the libraries MDAnalysis[73,74] and GetContacts (https://getcontacts.github.io/). To calculate relative positions of the inositol ring, we rotated and translated the trajectory by aligning Cα atoms of TM residues in proximity to the lipid-binding site and used the COM of the corresponding atoms as a reference. The binding poses were clustered using the gromos algorithm[35] based on the C atoms in the inositol ring using a cutoff of 0.1 nm.

### ATPase activity

The rate of ATP hydrolysis was monitored continuously using an enzyme-coupled assay, coupling ATP hydrolysis to a drop in NADH absorption at 340 nm. ATPase activity was measured at 37 °C in buffer C supplemented with 1 mM ATP, 1 mM phosphoenolpyruvate, $0.04\ mg.mL^{-1}$ pyruvate kinase, $0.1\ mg.mL^{-1}$ lactate dehydrogenase, 370 μM NADH, $1\ mg.mL^{-1}$ DDM (2 mM), $0.025\ mg.mL^{-1}\ PI(3,4,5)P_3$ and variable amounts of other phospholipids. In these experiments, the purified C-terminally truncated ATP8B1-CDC50A complex was added at a final concentration $2.3\ \mu g.mL^{-1}$. The NADH absorption decay was measured in a 96-well plate with a SpectraMax®i3 microplate reader in kinetic mode. Conversion from NADH oxidation rates expressed in $mAU.s^{-1}$ to ATPase activities expressed in $\mu mol.min^{-1}.mg^{-1}$ was based on the extinction coefficient of NADH at 340 nm.

### Reporting summary

Further information on research design is available in the Nature Portfolio Reporting Summary linked to this article.

## Data availability

Cryo-EM density maps have been deposited in the Electron Microscopy Data Bank under the accession codes EMD-17256 (E1-ATP), EMD-17257 (E1P-ADP), EMD-17258 (E1P), EMD-17259 (E2P$_{autoinhibited}$ "closed"), EMD-17260 (E2P$_{autoinhibited}$ "open"), EMD-17261 (E2P$_{active}$), EMD-17262 (E2⋯Pi (PS)), EMD-17262 (E2⋯Pi (PC)) and EMD-17263 (E2⋯Pi (PI)). Atomic coordinates have been deposited in the Protein Data Bank under IDs 8OX4 [https://doi.org/10.2210/pdb8OX4/pdb] (E1-ATP), 8OX5 [https://doi.org/10.2210/pdb8OX5/pdb] (E1P-ADP), 8OX6 [https://doi.org/10.2210/pdb8OX6/pdb] (E1P), 8OX7 [https://doi.org/10.2210/pdb8OX7/pdb] (E2P$_{autoinhibited}$ "closed"), 8OX8 [https://doi.org/10.2210/pdb8OX8/pdb] (E2P$_{autoinhibited}$ "open"), 8OX9 [https://doi.org/10.2210/pdb8OX9/pdb] (E2P$_{active}$), 8OXA [https://doi.org/10.2210/pdb8OXA/pdb] (E2⋯Pi (PS)), 8OXA (E2⋯Pi (PC)) and 8OXC [https://doi.org/10.2210/pdb8OXC/pdb] (E2⋯Pi (PI)). MD data and analysis have been deposited to the University of Copenhagen Electronic Research Data Archive (ERDA) and are available from: https://doi.org/10.17894/ucph.44e191c6-97ad-43ef-944f-be2ad95329fd. The data underlying Figs. 4a, 6b, and 7c are provided as a Source Data file. Source data are provided with this paper.

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

## Acknowledgements

We thank Thomas Boesen, Andreas Bøggild, and Taner Drace for technical support during EM data collection at the EMBION Danish National cryo-EM facility of Aarhus University (5072-00025B, Danish Agency for Research and Higher Education) and Jesper Lykkegaard Karlsen for scientific computing support. We are grateful to Anna Marie Nielsen, Tanja Klymchuk, and Bente Andersen for their technical assistance. We also wish to thank, Tomás Heger, Line Marie Christiansen, Ronja Driller, Cédric Montigny, Christine Jaxel, and David Stokes for discussion and advice. T.D. also warmly thank Camille and Côme for their patience during the revision process. This work was supported by a Marie Skłodowska-Curie Actions Individual Fellowship — LivFlip (101024542) and by a Fondation Recherche Médicale (ARF202209015714) grant to T.D., by Engineering and Physical Sciences Research Council grants (EP/X035603 and EP/V030779) to S. K., by an ANR grant (ANR-14-CE09-0022) to G.L., by the French Infrastructure for Integrated Structural Biology (FRISBI; ANR-10-INSB-05) to G.L., by the Center National de la Recherche Scientifique (CNRS) to G.L., by a Lundbeckfonden Fellow grant (R335-2019-2053) to J.A.L, by the Lundbeck Foundation to the BRAINSTRUC structural biology initiative (155-2015-2666) to K.L.-L. and P.N., and by a Lundbeckfonden Professorship grant (R310-2018-3713) to P.N.

## Author contributions

T.D. and P.N. conceived the project. T.D., F.K., K.L.-L., and P.N. designed the research. T.D. and F.K. performed the experiments. T.D., F.K., M.J.L, C.S., R.K.F., S.K., G.L., J.A.L., K.L.-L., and P.N. analyzed the data. T.D. drafted the manuscript. All authors contributed to the manuscript.

## Competing interests

The authors declare no competing interests.
