## [Peer Review File · Nature Communications]

Activation and substrate specificity of the human P4-ATPase ATP8B1REVIEWER COMMENTS

Reviewer #1 (Remarks to the Author):

The work by Dieudonne et al. is a follow-up of the study published by Dieudonne et al. in *Elife* (2022 Apr 13;11:e75272. doi: 10.7554/eLife.75272.). Whereas the *Elife* study resolved the cryo-EM structure of ATP8B1-CDC50A complex in the auto-inhibited (E2P) state, and showed auto-inhibition of the pump by C- and N-terminus as well as the requirement for phosphatidylinositols (in particular PIP3) in the activation of the pump, the present manuscript describes findings beyond this.

Firstly, the authors describe 9 different cryo-EM structures of the ATP8B1-CDC50A complex corresponding to different conformations in the Post-Albers catalytic cycle. In addition, the authors have identified and characterized the binding site for PIP3 in ATP8B1 as well as the binding pocket for different phospholipid headgroups/species; ATPase activity of the ATP8B1 was activated by the most common glycerophospholipids, including the novel substrate PI. A novel and interesting observation is the requirement of an ester bond at the sn-2 position of the glycerophospholipid for activation of ATPase activity of the protein.

I do have some suggestions to be addressed by the authors.

Results

1. In the ATPase assay (Fig. 5B) it is unclear what the source of the glycerophospholipids is. Please provide info like acyl chain length, saturation, source, manufacturer.
2. In line 297-298 the authors state: "In contrast, ether-PC, which lacks the fatty acid in sn-2 position, failed to efficiently activate ATP8B1". This is not correct as ether-PC has 2 acyl chains connected via an ether bond to the glycerol backbone? This is what the cartoon in Fig. 6C at least indicates.
3. The authors conclude (line 301-302): "Together, our data reveal that the ester bond in the sn-2 position is a critical determinant of lipid recognition by ATP8B1."

This conclusion is in my view too preliminary:

- a. One control ligand in Fig. 6 is missing for this conclusion to be valid: an ester bond on the sn-1 and an ether bond on the sn-2 position.
- b. The authors should also consider the number of acyl chains as possible determinants of lipid recognition and ATPase activation as lyso-PC does not inflict ATPase activation. From the experiments in Fig. 6, it could also be concluded that efficient substrate recognition requires 2 acyl chains on sn-1 and sn-2, from which one has to be connected to the glycerol backbone through an ester bond.

Discussion

1. The contribution of PIP3 to the activation of ATP8B1 is interesting and the authors could speculate on how they envision the interaction PIP3 and the phospholipid substrate with ATP8B1 binding pocket, i.e. are both lipids within the binding pocket at the same time? Is PIP3 released upon binding of substrate? How is PIP3 involved as an activator?

2. The discussion may benefit from published examples on associations between P4 ATPases and phosphoinositide metabolism.

Phosphoinositides are crucial regulators of membrane and protein trafficking events, predominantly via mobilization and activation of proteins of the trafficking machinery, i.e. PI4P is required for mobilization of the clathrin coat protein machinery to the trans Golgi Network (Wang YJ et al. *Cell* 114 (2003), 299; Graham TR and Burd CG *Trends Cell Biol* 21 (2011), 113). In relation to P4 ATPases, Puts et al. (*J. Proteome Res.* 9 (2010), 833) previously identified multiple Drs2p-interacting proteins involved in membrane trafficking, including three proteins with a role in phosphoinositide metabolism (including PI4P), underscoring the relevance of phosphoinositides in P4 ATPase function.

Similarly, the authors could speculate on the spatial distribution of PIP3 in relation to activation of P4 ATPase/ATP8B1 and cellular function.

Reviewer #2 (Remarks to the Author):

Dieudonné et al. determined a total of 9 cryo-EM structures of the human lipid flippase ATP8B1-CDC50A to provide insights into the mechanisms by which P4-ATPases transport phospholipids. In recent years, there have been many structural studies of P4-ATPases, namely human ATP8A1-CDC50A (*Science* 2019), human ATP11C-CDC50A (*Cell Rep* 2020; *JBC* 2020; *JBC* 2022), yeast Dnf1/2-Lem3 (*eLife* 2020; *Cell Report* 2022), yeast Drs2-Cdc50 (*Nature* 2019; *Nat Commun* 2019; *JMB* 2021), and ATP8B1-CDC50A (*eLife* 2022; *PNAS* 2022). Despite these available data, several aspects of the flippase mechanism remain incompletely understood, in part because many of these studies presented rather incomplete structural snapshots of the highly dynamic conformational cycles of these enzymes in piecemeal fashion. In this regard, the current manuscript by Dieudonné et al. perhaps provides the most complete picture so far with respect to the catalytic cycle (both active and autoinhibited states) of a P4-ATPase, which certainly offers new mechanistic insights. This includes three structures each in complex with a different phospholipid and the interaction with the sn-2 ester bond of substrate lipids. The quality of cryo-EM structures appears to be good overall based on figures and considering that they could identify water molecules in the transport active site. However, weaknesses are the fact that not many observations are completely novel (given the plethora of other P4-ATPases structures including the same authors' recent

paper of autoinhibited ATP8B1 in eLife) and unclear mechanistic significance of certain observations and conclusions. Overall, the quality and novelty of this manuscript would align with the caliber of Nature Communications. Nonetheless, several parts need revision before publication.

Major points:

1. The presence of water molecules in the lipid binding site is interesting, but the mechanistic importance of this is unclear. The current report is simply observational in this aspect, although it is understandable that this kind of hypothesis is difficult to test experimentally. Could these water molecules be simply passively occupying the hydrophilic space in the lipid binding site whenever it is available? Fig 5h seems compatible with such an idea, but the authors go beyond this with many speculations(L273-276). Maybe the authors can use MD simulations to support their model.

2. PI(3,4,5)P3-mediated activation is an important aspect of this flippase as shown in Fig 4a. Unfortunately, the current structural data does not provide insight into the mechanism because the authors did not observe conformational changes throughout the structures of different catalytic states. This raises a concern whether the identified PI(3,4,5)P3 binding site is indeed the one that mediates activation. This issue can be addressed by a structure without PI(3,4,5)P3 and/or by mutational analysis (measuring PI(3,4,5)P3-dependent activation).

3. Six out of the nine cryo-EM structures are obtained with PI(4,5)P2 instead of PI(3,4,5)P3. Yet, there is no mention about PI(4,5)P2 in the manuscript and what are differences between the two phosphoinositides in terms of the ATPases activation and the structure. This must be addressed in the text.

4. MD simulations on PI(3,4,5)P3: It is confusing what is actually observed here (L200-205). I guess “remained bound (L202)” and “stable binding (L203)” could be the reason because readers would get an impression that the binding is through defined interactions with the protein. Does the current description mean that both PI(3,4,5)P3 and PI stay in the pocket during the duration of simulations, but their positions (such as headgroup orientations) are highly dynamic with respect to the pocket? Then, I am not sure how to reconcile this observation with the more or less defined position of PI(3,4,5)P3 in the cryo-EM structure (also a statement like “PI(3,4,5)P3 is the most well-ordered ligand”; L236), unless at least cluster 1 (57%) is the same structure as in the cryo-EM structure.

5. Substrate lipid-bound structures: Based on structures, all tested lipids PC, PS, and PI seem to bind to the pocket without a problem (and without inducing a significant change in the pocket). Yet, there is quite a significant difference in ATPase stimulation between PC vs PI/PS. This requires additional analysis and discussion to provide plausible explanations. For example, would PS or PI sterically block

conformational changes during the E2-Pi to E2 transition? Additionally, the authors should also include additional density/model overlay figures to show that the models are convincing (difficult to judge from the current figures) or provide additional evidence supporting that the observed lipid densities are indeed the modeled lipids. Almost all membrane proteins co-purify with endogenous lipids as a mixed detergent/lipid micelle, and it is possible that co-purified PC in the micelle preferentially binds back to the pocket despite the supply of exogenous lipids.

6. Figure 6 a and b: I do not understand why there are three rotamers of S403 overlaid here. Is the S403 sidechain clearly resolved in the cryo-EM maps? If not, these models would be too speculative. I am not sure how the water molecule can be defined in a single position whereas the S403 sidechain is not. How do S403 mutations affect the activity of ATP8B1 (or other P4-ATPases if known)?

Minor points

L78: As the authors mentioned later in the text, the current evidence suggests that substrate transport is also linked to dephosphorylation in the cases of both P5A and P5B ATPases. Thus, “unique to P4-ATPases” needs to be corrected.

L88: strange insertion of a date/time.

Fig 2c. In the E1p-ADP structure, the sidechain OH group is pointing in a different direction compared to the other two structures. How accurate is the rotamer assignment and is this a real difference?

Fig 2 legend: need the unit (σ) for contour map levels.

Fig 3 and Supplementary Figs 7 and 9: need a scale bar for surface electrostatic maps.

Fig 5f: Given the resolution, I am not convinced whether all modelled water molecules could be individually resolved. This needs some descriptions in the legend as well as an additional supplementary figure.

L170: the authors should provide a definite number rather than “a significant fraction.”

L195: “partial occupancy” instead of “heterogeneous occupation”?

Fig 4e,f,h: mark the position corresponding to the cryo-EM structure.

Methods: Detailed information (steps and particle numbers) about cryo-EM data processing procedures for each sample should be included. Additional schematics would be helpful if the procedures deviated from the one shown in Supplementary Fig S3.

Supplementary Fig S3: some fonts in the FSC and particle orientation charts are illegible.

Supplementary Fig S6: Indicate RMSDs.

Reviewer #3 (Remarks to the Author):

See PDF comments.

This is absolutely beautiful work. Dieudonné and colleagues present nine structures of the human flippase. These structures provide structural insight on the function of these biomolecular machines. In particular, the authors provide structural insight into a water molecule that is bound at the canonical ion binding site. Furthermore, the authors investigated the protein-lipid interactions that were observed in the density using molecular dynamics (MD) simulations. They found that PI(3,4,5)P3 was able to form more stable contacts with the protein than its PI counterpart. Overall, the manuscript is well-written and significantly contributes to the understanding of transport of cargo through this P-type Pump. I have several comments and questions that I hope the authors will be able to address.

The text says (line 108) “Here, we present nine different cryo-EM structures, functional studies, and molecular dynamics simulation data of the ATP8B1-CDC50A complex in conformations covering most of the transport cycle and different substrate complexes.” True enough and impressive structural work all around. However, when I look at all the structures displayed in Figure 1, I am confused by the purported mechanism. All the structures shown, the bound lipid is in an orientation that is consistent with the outer leaflet (polar head up and carbon chains down). So this is a flippase, but no lipid is shown flipped. I am mentally trying to picture the complete transport cycle from these pictures. The flippases actively transport lipids from the outer leaflet to the inner leaflet. Thus, to complete the transport cycle and release the bound lipid to the inner leaflet, should one also expect to see (at least transiently) a lipid upside down (polar head up and carbon chains down) before it is released to the inner leaflet? Why isn't there any structure with a bound lipid upside down? Is this intermediate state so short-lived that it cannot be captured experimentally? Should this be discussed?

The text says (line 356-358) “...questions remain open on how phosphoinositides activate ATP8B1 and potentially interfere with autoinhibition, and which specific steps of the catalytic cycle are stimulated. “ It was not immediately clear to me how the binding of PI(3,4,5)P3 relates to the TM1-TM2 responsible for autoinhibition. I tried to compare Figure 1 and Figure 4 to understand the relationship, but it is not easy. In the context of Figure 4 it might be helpful to add a view where both the PI(3,4,5)P3 binding site and of a lipid substrate are visualized clearly. Based on these two pictures, they seem to be on opposite sides of the TM1-TM2 involved in the autoinhibition. Also could more be said about how the PI(3,4,5)P3 affects the TM1-TM2 mechanistically?

There is density attributed to a water molecule located in the canonical transport site of P-type ATPase in the E1 states of the cycle. This position partially overlaps with heavy metals and cation sites of P1 and P2-ATPases. The authors are excited by this observation. While I realize that this has not been observed previously, the presence of a water near a highly polar (potentially for binding an ion or a lipid polar head) should have been expected. What happened then? Does this mean that water was undetected previously but was perhaps present, or that this location was truly dehydrated in these other structures? Was the absence of water at this location in these other structures questioned previously? Please discuss.

The observed density of the second magnesium ion in the E1-ATP structure (Figure 1) deserves some comments. As far as I recall, there is no other structure complexed with AMPPCP that observes this second density—however—computational studies have suggested that there might be a second catalytic magnesium (see Mateeva et al, 2021, *J. Phys. Chem. B*, 125, 11835, who modelled a P-type ATPase with two and one magnesium and proposed that two Mg are required). Other structures of P-type pumps observe a second magnesium but only in the presence of AlF₄⁻. Insight regarding the second magnesium would be illuminating.

The text says that “Regulation through N- and C-terminal tails is a feature shared among various P-type ATPases, but structurally it was first described for the Drs2p-Cdc50p flippase complex, which revealed how the C-terminal tail occupies the nucleotide binding pocket and locks the cytoplasmic domains movements.” Still, I am confused about the functional significance of the autoinhibited ATP8B1-CDC50A conformation(s). Is the occurrence of this conformation submitted to some regulatory signal? If so, what is it? If not, then why are these conformations considered “off-cycle” rather than part of the transport cycle associated? It seems that TM1 and TM2 play the role of some gate that can open and close the lipid binding site.

Minor point:

The text says (line 78) “Unique to P4-ATPases, the lipid transport is linked to the dephosphorylation reaction...” Isn't this like the K⁺ ions in the Na/K ATPase (they are released upon dephosphorylation)?

On page 4, it might be helpful to clarify that the CDC50A extracellular domain is required for forming a functional complex with and chaperoning phospholipid flippases to the plasma membrane [Segawa et al, 2018, *Cell Biol* 293(6), 2172].

The text says (line 112-113) "We also report that the autoinhibited ATP8B1-CDC50A complex oscillates between an outward-open and ..." I recommend to replace the word oscillates by fluctuates.

RESPONSE TO THE REVIEWERS

Reviewer #1 (Remarks to the Author):

The work by Dieudonne et al. is a follow-up of the study published by Dieudonne et al. in Elife (2022 Apr 13;11:e75272. doi: 10.7554/eLife.75272.). Whereas the Elife study resolved the cryo-EM structure of ATP8B1-CDC50A complex in the auto-inhibited (E2P) state, and showed auto-inhibition of the pump by C- and N-terminus as well as the requirement for phosphatidylinositols (in particular PIP3) in the activation of the pump, the present manuscript describes findings beyond this.

Firstly, the authors describe 9 different cryo-EM structures of the ATP8B1-CDC50A complex corresponding to different conformations in the Post-Albers catalytic cycle. In addition, the authors have identified and characterized the binding site for PIP3 in ATP8B1 as well as the binding pocket for different phospholipid headgroups/species; ATPase activity of the ATP8B1 was activated by the most common glycerophospholipids, including the novel substrate PI. A novel and interesting observation is the requirement of an ester bond at the sn-2 position of the glycerophospholipid for activation of ATPase activity of the protein.

We thank the Reviewer for their appreciation of our work.

I do have some suggestions to be addressed by the authors.

Results

1. In the ATPase assay (Fig. 5B) it is unclear what the source of the glycerophospholipids is. Please provide info like acyl chain length, saturation, source, manufacturer.

A new "Lipids" paragraph has been now added to the Methods section. This section now contains all the information about the lipids used in this study.

2. In line 297-298 the authors state: "In contrast, ether-PC, which lacks the fatty acid in sn-2 position, failed to efficiently activate ATP8B1". This is not correct as ether-PC has 2 acyl chains connected via an ether bond to the glycerol backbone? This is what the cartoon in Fig. 6C at least indicates.

We thank the Reviewer for this comment as it highlights a mistake. Indeed, the Reviewer is right, ether-PC contains two acyl chains, both connected with an ether bond to the glycerol backbone. The text has been now corrected.

3. The authors conclude (line 301-302): "Together, our data reveal that the ester bond in the sn-2 position is a critical determinant of lipid recognition by ATP8B1."

This conclusion is in my view too preliminary:

a. One control ligand in Fig. 6 is missing for this conclusion to be valid: an ester bond on the sn-1 and an ether bond on the sn-2 position.

We agree with the reviewer that this control ligand would reinforce our point. Unfortunately, this non-natural lipid (sn-2 mono ether PC) is not commercially available and would require some dedicated chemical synthesis and a project on its own. We have also modified our conclusions and discussion to take into account this particular point.

b. The authors should also consider the number of acyl chains as possible determinants of lipid recognition and ATPase activation as lyso-PC does not inflict ATPase activation. From the experiments in Fig. 6, it could also be concluded that efficient substrate recognition requires 2 acyl chains on sn-1 and sn-2, from which one has to be connected to the glycerol backbone through an ester bond.

We agree with the reviewer that our data indicate that the number of acyl chains is another influential factor in substrate recognition. As a result, we have made modifications to the results and discussion sections to incorporate this possibility arising from our findings.

Discussion

1. The contribution of PIP3 to the activation of ATP8B1 is interesting and the authors could speculate on how they envision the interaction PIP3 and the phospholipid substrate with ATP8B1 binding pocket, i.e. are both lipids within the binding pocket at the same time?

We thank the Reviewer for these comments that highlight gaps in our explanation of results that would benefit from an additional figure. The PI(3,4,5)P₃ binding site is distinct from the phospholipid binding/transport site. Indeed the identified PI(3,4,5)P₃ binding site is located in between the last transmembrane segments of ATP8B1 on the cytosolic side of the membrane, whereas the phospholipid transport site results from the opening of TM1-2, also on the cytosolic leaflet. To clarify this point, Figure 4 was modified to include an overall view of the ATP8B1-CDC50A complex in E2P_{active} conformation where both sites are filled with their respective lipids.

Is PIP3 released upon binding of substrate?

Our data tends to indicate that PI(3,4,5)P₃ is not released upon binding of substrate as a nearly identical conformation of PI(3,4,5)P₃ was observed in the three E2P conformations, including the E2P_{active} conformation which is bound to PC (Supplementary Figures 16a). Additionally, the comparison of the TM organization in all conformations of ATP8B1 in this study shows that the site would be able to accommodate PI(3,4,5)P₃ at different step of its catalytic cycle (Supplementary Figure 19a).

How is PIP3 involved as an activator?

This is a key question, but unfortunately, we cannot conclude strongly on this point from our data except pointing to possible mechanisms (see discussion). More work on this direction will be needed to better understand the physiological relevance and

molecular basis of this activation. The discussion has been modified to further discuss this.

2. The discussion may benefit from published examples on associations between P4 ATPases and phosphoinositide metabolism.

Phosphoinositides are crucial regulators of membrane and protein trafficking events, predominantly via mobilization and activation of proteins of the trafficking machinery, i.e. PI4P is required for mobilization of the clathrin coat protein machinery to the trans Golgi Network (Wang YJ et al. Cell 114 (2003), 299; Graham TR and Burd CG Trends Cell Biol 21 (2011), 113). In relation to P4 ATPases, Puts et al. (J. Proteome Res. 9 (2010), 833) previously identified multiple Drs2p-interacting proteins involved in membrane trafficking, including three proteins with a role in phosphoinositide metabolism (including PI4P), underscoring the relevance of phosphoinositides in P4 ATPase function. Similarly, the authors could speculate on the spatial distribution of PIP3 in relation to activation of P4 ATPase/ATP8B1 and cellular function.

We thank the reviewer for this point. We have now updated our discussion to further highlight the role of phosphoinositides in lipid homeostasis and reported interactions with other P-type ATPases. The spatial distribution of phosphoinositides in relation to activation of P4-ATPase/ATP8B1 and cellular function was also discussed in our previous study (Dieudonné et al., eLife 2022).

Reviewer #2 (Remarks to the Author):

Dieudonné et al. determined a total of 9 cryo-EM structures of the human lipid flippase ATP8B1-CDC50A to provide insights into the mechanisms by which P4-ATPases transport phospholipids. In recent years, there have been many structural studies of P4-ATPases, namely human ATP8A1-CDC50A (Science 2019), human ATP11C-CDC50A (Cell Rep 2020; JBC 2020; JBC 2022), yeast Dnf1/2-Lem3 (eLife 2020; Cell Report 2022), yeast Drs2-Cdc50 (Nature 2019; Nat Commun 2019; JMB 2021), and ATP8B1-CDC50A (eLife 2022; PNAS 2022). Despite these available data, several aspects of the flippase mechanism remain incompletely understood, in part because many of these studies presented rather incomplete structural snapshots of the highly dynamic conformational cycles of these enzymes in piecemeal fashion. In this regard, the current manuscript by Dieudonné et al. perhaps provides the most complete picture so far with respect to the catalytic cycle (both active and autoinhibited states) of a P4-ATPase, which certainly offers new mechanistic insights. This includes three structures each in complex with a different phospholipid and the interaction with the sn-2 ester bond of substrate lipids. The quality of cryo-EM structures appears to be good overall based on figures and considering that they could identify water molecules in the transport active site. However, weaknesses are the fact that not many observations are completely novel (given the plethora of other P4-ATPases structures including the same authors' recent paper of autoinhibited ATP8B1 in eLife) and unclear mechanistic significance of certain observations and conclusions. Overall, the quality and novelty of this manuscript would align with the caliber of Nature Communications. Nonetheless, several parts need revision before publication.

We thank the Reviewer for their appreciation of our work.

Major points:

1. The presence of water molecules in the lipid binding site is interesting, but the mechanistic importance of this is unclear. The current report is simply observational in this aspect, although it is understandable that this kind of hypothesis is difficult to test experimentally. Could these water molecules be simply passively occupying the hydrophilic space in the lipid binding site whenever it is available? Fig 5h seems compatible with such an idea, but the authors go beyond this with many speculations (L273-276). Maybe the authors can use MD simulations to support their model.

We appreciate the Reviewer comments and suggestions. We acknowledge that our structural data are observational and do not provide mechanistic insights into the role of water molecules in the lipid occlusion site of ATP8B1. As mentioned by the reviewer, testing this hypothesis experimentally would be challenging, and similar studies conducted on the Drs2p-Cdc50p complex required extensive work using MD simulations (<https://www.biorxiv.org/content/10.1101/2020.06.24.169771v2>). While we agree that in silico experiments might offer valuable information on the occlusion mechanism of lipids by ATP8B1, we believe that exploring this aspect, requiring many variations and controls, is another project on its own and beyond the scope of this manuscript, which reports on the mechanism of substrate recognition and phosphoinositide binding. We have revised the results and discussion section to provide a clearer emphasis of our findings and also reduce the speculative part.

2. PI(3,4,5)P₃-mediated activation is an important aspect of this flippase as shown in Fig 4a. Unfortunately, the current structural data does not provide insight into the mechanism because the authors did not observe conformational changes throughout the structures of different catalytic states. This raises a concern whether the identified PI(3,4,5)P₃ binding site is indeed the one that mediates activation. This issue can be addressed by a structure without PI(3,4,5)P₃ and/or by mutational analysis (measuring PI(3,4,5)P₃-dependent activation).

We agree that our data provide neither a straightforward model for the activation mechanism of ATP8B1 by PI(3,4,5)P₃ nor for the regulation of this activation in vivo (see also above). We hoped that our structural findings would have lead to clearer clues, but they do not, except pointing to the phosphoinositide binding site and indicating differences in flexibility between PI and PI(3,4,5)P₃. We do not think that another PI(3,4,5)P₃ site is involved, but we cannot exclude it fully, of course. We have modified the discussion to emphasize these points, which should hopefully inspire future studies.

As mentioned by Reviewer 2, structural and functional experiments combined with in silico studies on ATP8B1 (and Drs2p) is likely to improve our understanding of which specific part of the catalytic cycle is limited by the lack of phosphoinositides in P4-ATPases, but it goes beyond the current study.

3. Six out of the nine cryo-EM structures are obtained with PI(4,5)P₂ instead of PI(3,4,5)P₃. Yet, there is no mention about PI(4,5)P₂ in the manuscript and what are

differences between the two phosphoinositides in terms of the ATPases activation and the structure. This must be addressed in the text.

The result and the discussion sections has been modified to include this information. PI(4,5)P₂ also activates ATP8B1 and has a significant bioavailability, as discussed also in our earlier paper (Dieudonné et al., eLife 2022).

4. MD simulations on PI(3,4,5)P₃: It is confusing what is actually observed here (L200-205). I guess “remained bound (L202)” and “stable binding (L203)” could be the reason because readers would get an impression that the binding is through defined interactions with the protein. Does the current description mean that both PI(3,4,5)P₃ and PI stay in the pocket during the duration of simulations, but their positions (such as headgroup orientations) are highly dynamic with respect to the pocket? Then, I am not sure how to reconcile this observation with the more or less defined position of PI(3,4,5)P₃ in the cryo-EM structure (also a statement like “PI(3,4,5)P₃ is the most well-ordered ligand”; L236), unless at least cluster 1 (57%) is the same structure as in the cryo-EM structure.

We thank the reviewer for pointing out inconsistencies in our description. We have modified the results section to clarify our findings. Both PI and PI(3,4,5)P₃ were observed to remain in the pocket throughout the simulations. We suggest that the observed difference in the position of PI(3,4,5)P₃ between the cryo-EM data and the MD simulation may result from various factors: 1) MD simulations were performed in a membrane with a composition mimicking the plasma membrane, while the cryo-EM data were obtained in LMNG mixed micelles. The difference in the surrounding lipid environment could contribute to variations. 2) MD simulations were conducted at a constant temperature of 37 °C, whereas the sample used for cryo-EM was prepared at 4°C and flash-cooled. The disparity in temperature conditions could influence the conformational behavior of PI(3,4,5)P₃. 3) The cryo-EM density likely reflect an ensemble of PI(3,4,5)P₃ conformations and therefore results in a less defined density compared to the surrounding protein pocket, albeit better defined than for other phosphoinositides. This results in limited precision in modeling and refinement of PI(3,4,5)P₃ in our atomic model, but this is best we can obtain. To emphasize this point, and as ask by the Reviewer, we have modified the Figure 5 b,c and e to illustrate the position of PI(3,4,5)P₃ as modeled in our cryo-EM data and observed during MD simulations.

5. Substrate lipid-bound structures: Based on structures, all tested lipids PC, PS, and PI seem to bind to the pocket without a problem (and without inducing a significant change in the pocket). Yet, there is quite a significant difference in ATPase stimulation between PC vs PI/PS. This requires additional analysis and discussion to provide plausible explanations. For example, would PS or PI sterically block conformational changes during the E2-Pi to E2 transition?

We appreciate the Reviewer's comment, and we have made the necessary modifications to the manuscript to address this point. However, while our data demonstrates that PC/PS/PI can be occluded within the protein without affecting the occlusion site geometry, we cannot draw conclusions about the likelihood and kinetics

of these lipids reaching the lipid transport site when the groove is open in the E2P active conformations. The obvious explanation will be that the differences observed in our ATPase activity measurements are due to variations in the speed of lipid access to the site, but going further on this model will require very significant resources and time with MD simulations. Therefore, we conclude here that the various lipids are substrates (based on ATPase stimulation) and can be occluded at a consistent site.

Furthermore, our data suggests that the observed differences also may relate to variations in water network associated with lipid head group occlusion. As the reviewer suggested, it is possible that the different lipid head groups may not cross the "hydrophobic gate" formed by I402 during the E2-Pi to E2 transition at the same speed, and therefore would affect ATPase activity as suggested by the reviewer. However, as discussed with Reviewer 1, our data do not provide structural information regarding the lipid translocation that occurs during the E2-Pi to E2 transition, and MD simulation of these changes require far more analysis than can be covered for this report. We therefore decided not to speculate further in this report.

Additionally, the authors should also include additional density/model overlay figures to show that the models are convincing (difficult to judge from the current figures) or provide additional evidence supporting that the observed lipid densities are indeed the modeled lipids.

We have now included new figures (Supplemental Figure 18 and Supplemental movie 1) to providing more detailed view of the lipid occlusion site and its associated density in the E2-Pi (PC), E2-Pi (PS), E2-Pi (PI) conformations.

Almost all membrane proteins co-purify with endogenous lipids as a mixed detergent/lipid micelle, and it is possible that co-purified PC in the micelle preferentially binds back to the pocket despite the supply of exogenous lipids.

We agree on this point. Similarly, other P4-ATPase structures obtained by cryo-EM were captured in a E2-Pi transition state by the phosphate mimic AlF_x , even in the trace amounts of lipids, e.g. Drs2p E1P-ADP and E2-Pi(PS) derived from the same EM sample and data set (Timcenko et al., 2021), and ATP8A1 E1P and E2-Pi(PS) also derived from the same sample and data (Hiraizumi et al., 2019).

This has been a major concern, when designing experiments to obtain the E2-Pi(PC/PS/PI) structures. Washing steps were extensively prolonged and the concentration of detergent increased to promote the slow exchange of endogenous lipids with detergent micelles (Montigny et al., 2017). The Methods section concerning the sample purification has been modified to emphasize this point.

To ensure that lipid exchange was complete and EM density represented the exogenous lipids added, we conducted also cryo-EM control experiments in the presence of AlF_x with the exact same protein sample as used for the E2-Pi (PS) and E2-Pi (PI) data sets, but in absence of exogenous lipids. As described in the new Supplementary Figure 5, for this experiment (AlF_x only), a large data set (10,640 micrograph) was collected leading to a stack of 813,354 of particles after 2D/3D

sorting. Despite many different approaches (Heterologous refinement, 3D classification without alignment, 3D variability), no E2-Pi like conformation was observed in these data sets.

In contrast, when the sample is supplemented with exogenous lipid and vanadate (vanadate stabilizes the same E2-Pi transition conformation as AlF_x) we obtained a very homogenous particle stack corresponding to a lipid-occluded E2-Pi like conformation. For these reasons, we are confident that the lipid observed in the occlusion site is not a co-purified lipid, but corresponds to the pure exogenous lipid species added before grid freezing, and our density maps fully support this.

6. Figure 6 a and b: I do not understand why there are three rotamers of S403 overlaid here. Is the S403 sidechain clearly resolved in the cryo-EM maps? If not, these models would be too speculative. I am not sure how the water molecule can be defined in a single position whereas the S403 sidechain is not.

We thank the Reviewer for these comments. Our intention is only to indicate the possibility for all S403 rotamers to make productive interactions with the lipid backbone, but we have made a change in our Figure 8 (previously 7a) and a new Supplemental Figure 18 to be more clear. For the E2P_{active} conformation, the S403 sidechain is clearly resolved, and the cryo-EM density also shows a clear interaction with a water molecule. However, for the occluded E2-Pi conformations (with PC, PS, or PI, see above), no density can be observed for the water molecule, and the density for the S403 sidechain indicates flexibility, i.e. the presence of different rotamers, including the rotamers p and t, which would interact with different parts of the lipid. For all three possible rotamers, they can interact productively by hydrogen bonding. We have included a new Supplemental Figure 18 that provides a clear view of the density surrounding S403 in the aforementioned conformations. Additionally, we modified the panel a of the main Figure 8 (previously 7a) to remove the display of S403 rotamers in the E2P active conformation, where the side chain indeed appears fixed on a specific interaction.

How do S403 mutations affect the activity of ATP8B1 (or other P4-ATPases if known)?

The S403Y mutation in ATP8B1 has been reported as one of the mutations leading to the most severe form of inherited intrahepatic cholestasis known as Progressive Familial Intrahepatic Cholestasis type 1 (PFIC1) (Klomp et al., 2004). Additionally, the S403Y mutation has been shown to dramatically reduce the ATPase activity, by approx. 85%, of the purified protein (Chen et al., 2022). We thank the reviewer for this comment and the results section has been modified to include this information in the manuscript.

Additionally, the effect of a more conservative mutation (S365A) has been investigated by Vestergaard and colleagues on the PS/PE specific P4-ATPase, ATP8A2 (<https://www.pnas.org/doi/10.1073/pnas.1321165111>). They could observe a decrease of the V_{max} of the enzyme of 25 % and 85% with PS and PE, respectively. Moreover, the mutation also reduces the apparent affinity of the enzyme by 2.7 and 2.3-fold for PS and PE, respectively.

Minor points

L78: As the authors mentioned later in the text, the current evidence suggests that substrate transport is also linked to dephosphorylation in the cases of both P5A and P5B ATPases. Thus, “unique to P4-ATPases” needs to be corrected.

Indeed, thanks. Corrected

L88: strange insertion of a date/time.

Corrected

Fig 2c. In the E1p-ADP structure, the sidechain OH group is pointing in a different direction compared to the other two structures. How accurate is the rotamer assignment and is this a real difference?

We thank the Reviewer for these comments. In the E1P-ADP cryo-EM map, the sidechain of residue 403 is not clearly resolved (see picture below) and both orientations of the side chain seem possible. However, as pointed out by the Reviewer, given the presence of the water molecule it's more likely that the S403 points toward the water molecule (distance 2.6Å). We therefore modified the S403 sidechain orientation and re-refined the model in Phenix. The new model was used to generate a new Figure 2c.

Fig 2 legend: need the unit (sigma) for contour map levels.

Corrected

Fig 3 and Supplementary Figs 7 and 9: need a scale bar for surface electrostatic maps.

Corrected

Fig 5f: Given the resolution, I am not convinced whether all modelled water

molecules could be individually resolved. This needs some descriptions in the legend as well as an additional supplementary figure.

We have now included a supplemental movie showing the lipid occlusion site of the models and their associated density maps.

L170: the authors should provide a definite number rather than “a significant fraction.”

Corrected

L195: “partial occupancy” instead of “heterogeneous occupation”?

Corrected

Fig 4e,f,h: mark the position corresponding to the cryo-EM structure.

Corrected

Methods: Detailed information (steps and particle numbers) about cryo-EM data processing procedures for each sample should be included. Additional schematics would be helpful if the procedures deviated from the one shown in Supplementary Fig S3. Supplementary Fig S3: some fonts in the FSC and particle orientation charts are illegible.

Individual Supplementary figures (Supp. Figure 3 to 10) have been added to provide more details about the cryo-EM data processing of each data set.

Supplementary Fig S6: Indicate RMSDs.

Corrected

Reviewer #3 (Remarks to the Author):

This is absolutely beautiful work. Dieudonné and colleagues present nine structures of the human flippase. These structures provide structural insight on the function of these biomolecular machines. In particular, the authors provide structural insight into a water molecule that is bound at the canonical ion binding site. Furthermore, the authors investigated the protein-lipid interactions that were observed in the density using molecular dynamics (MD) simulations. They found that PI(3,4,5)P3 was able to form more stable contacts with the protein than its PI counterpart. Overall, the manuscript is well-written and significantly contributes to the understanding of transport of cargo through this P-type Pump. I have several comments and questions that I hope the authors will be able to address.

We thank the Reviewer for his/her appreciation of our work.

The text says (line 108) “Here, we present nine different cryo-EM structures, functional studies, and molecular dynamics simulation data of the ATP8B1-CDC50A complex in conformations covering most of the transport cycle and different substrate complexes.” True enough and impressive structural work all around. However, when I look at all

the structures displayed in Figure 1, I am confused by the purported mechanism. All the structures shown, the bound lipid is in an orientation that is consistent with the outer leaflet (polar head up and carbon chains down). So this is a flippase, but no lipid is shown flipped. I am mentally trying to picture the complete transport cycle from these pictures. The flippases actively transport lipids from the outer leaflet to the inner leaflet. Thus, to complete the transport cycle and release the bound lipid to the inner leaflet, should one also expect to see (at least transiently) a lipid upside down (polar head up and carbon chains down) before it is released to the inner leaflet? Why isn't there any structure with a bound lipid upside down? Is this intermediate state so short-lived that it cannot be captured experimentally? Should this be discussed?

We thank the reviewer for raising this point. We acknowledge that our structures do not cover the complete transport cycle of ATP8B1, as we did not capture the final step where the lipid is translocated to the other side of the membrane. Upon translocation the acyl chains remain in the hydrophobic core of the membrane. The close-up view of the occluded lipids from the E2-Pi in Fig1 is not perpendicular to the overall view of the EM density (here the view was chosen to better show the lipid and its associated EM density).

From a structural point of view, when the lipid is occluded, the head group would need to pass a "hydrophobic gate" formed in part by the Ile402 of the TM4 PISL motif (originally described by Vestergaard et al., 2014). Despite employing various approaches, we were able to obtain neither the conformation with a lipid bound after crossing this hydrophobic gate, i.e. flipped to the other side of the membrane, nor gather structural information regarding how the lipid headgroup can pass through this "hydrophobic gate."

As the reviewer suspects, we also believe that the lipid release conformation is transient and with a low affinity for the lipid, which would explain the challenge in

capturing it. We have chosen not to discuss this point extensively in our study because it has been previously addressed in the research conducted by the Nureki's lab on ATP8A1 (Hiraizumi et al., 2019) and the Abe's lab on ATP11C (Nakanishi et al., 2020a & Nakanishi et al., 2020b) - and unfortunately, our study does not provide additional information on this particular aspect.

The text says (line 356-358) "...questions remain open on how phosphoinositides activate ATP8B1 and potentially interfere with autoinhibition, and which specific steps of the catalytic cycle are stimulated. " It was not immediately clear to me how the binding of PI(3,4,5)P₃ relates to the TM1-TM2 responsible for autoinhibition. I tried to compare Figure 1 and Figure 4 to understand the relationship, but it is not easy. In the context of Figure 4 it might be helpful to add a view where both the PI(3,4,5)P₃ binding site and of a lipid substrate are visualized clearly. Based on these two pictures, they seem to be on opposite sides of the TM1-TM2 involved in the autoinhibition.

Actually, TM1-2 does not necessarily relate to regulation of ATP8B1 by PI(3,4,5)P₃ and TM1-2 is not responsible for autoinhibition. The reviewer might have been confused by the lack of clarity of our manuscript. Therefore, and as also suggested by reviewer 1 we have now included a new Figure 4c displaying a global view of the ATP8B1-CDC50A complex in the E2P_{active} conformation with both sites filled (substrate site with PC and regulatory site with PI(3,4,5)P₃).

Also could more be said about how the PI(3,4,5)P₃ affects the TM1-TM2 mechanistically? PI(3,4,5)P₃ binding site PC lipid binding site.

Our results do not specifically point toward a model where PI(3,4,5)P₃ binding affects the structural organization and dynamics of TM1-TM2. It is important to note that from our structural data and based on the accumulated information from ATP8B1 and Drs2p, it is likely that the dynamics of the dephosphorylation phase of the ATPase cycle (which is associated with lipid transport) is facilitated by phosphoinositides. More extensive research is necessary to elucidate whether the activation is linked to the dynamics of opening of the TM1-2 region, a subtle but crucial reorganization of the cytosolic domains, or another as-yet-unknown mechanism. The discussion of the manuscript now has been revised to further emphasize this point and hopefully stimulate and inspire future experiments.

There is density attributed to a water molecule located in the canonical transport site of P-type ATPase in the E1 states of the cycle. This position partially overlaps with heavy metals and cation sites of P1 and P2-ATPases. The authors are excited by this observation. While I realize that this has not been observed previously, the presence of a water near a highly polar (potentially for binding an ion or a lipid polar head) should have been expected. What happened then? Does this mean that water was undetected previously but was perhaps present, or that this location was truly dehydrated in these other structures? Was the absence of water at this location in these other structures questioned previously? Please discuss.

We thank the reviewer for this comment. Indeed, the observation of this water molecule is not unexpected per se. However, we believe that this observation provides noteworthy information about the evolutionary relationships between P-type ATPase families and might facilitate further mechanistic understanding. Most likely, P4- and P5-ATPases evolved from cation transporters. In P4 and P5 the phosphorylation step has so far been considered “substrate-independent”, while our analysis seems to indicate that the “cation site” (found in the P1- and P2-ATPases) remains a polar site, which is then important for lipid backbone recognition along with the dephosphorylation half-cycle. For the E1 half-cycle, it simply evolved into a water molecule site for stabilization and stimulation of the phosphorylation. In Supplemental Figure 11, we identify this water molecule in previously published structures of P4-ATPases, where it had not been modelled. In P5-ATPases, the water molecules were modeled, but their presence was not further discussed and perhaps not recognized as a mechanistic feature (Sim et al., 2021). The discussion section has been modified to address our point more clearly.

The observed density of the second magnesium ion in the E1-ATP structure (Figure 1) deserves some comments. As far as I recall, there is no other structure complexed with AMPPCP that observes this second density—however—computational studies have suggested that there might be a second catalytic magnesium (see Mateeva et al, 2021, J. Phys. Chem. B, 125, 11835, who modelled a P-type ATPase with two and one magnesium and proposed that two Mg are required). Other structures of P-type pumps observe a second magnesium but only in the presence of AIF2. Insight regarding the second magnesium would be illuminating.

We thank the reviewer for addressing this relevant point. We have included an additional discussion on this specific point in our revised manuscript. As mentioned by the reviewer, our initial modeling included a second magnesium ion based on the observation of a second ion in other structures stabilized with AIF₄ and ADP (first noted by Sørensen et al. 2004, Science). However, after comparison of all cryo-EM data of P-type ATPases obtained in the presence of AMPPCP, and considering the in silico studies conducted by Mateeva and colleagues, we have reconsidered the assignment of this density. After careful consideration of all the gathered information, we have made the decision not to model any specific ion into the electron density due to the uncertainty surrounding the exact nature of the density.

The text says that “Regulation through N- and C-terminal tails is a feature shared among various P-type ATPases, but structurally it was first described for the Drs2p-Cdc50p flippase complex, which revealed how the C-terminal tail occupies the nucleotide binding pocket and locks the cytoplasmic domains movements.” Still, I am confused about the functional significance of the autoinhibited ATP8B1-CDC50A conformation(s). Is the occurrence of this conformation submitted to some regulatory signal? If so, what is it? If not, then why are these conformations considered “off-cycle” rather than part of the transport cycle associated? It seems that TM1 and TM2 play the role of some gate that can open and close the lipid binding site.

We thank the reviewer for this relevant question. We believe that the autoinhibition mechanism is a key regulatory mechanism for the flippase activity of ATP8B1, together

with phosphoinositides binding. In vitro, to obtain an active sample, we have to produced a full-length protein containing a 3C protease to cleave the C-terminal region by proteolysis to relieve its autoinhibition (genetic truncations impairs protein expression). However, as far as we are aware, there is no indication of such a mechanism to activate the protein in vivo. Therefore, we hypothesize that regulatory proteins might be able to bind to the N- and C-terminal tails of ATP8B1 to unleash the transport activity of the flippase, but these proteins have not been identified yet. As previously discussed in Dieudonné et al. 2020, post-translational modifications such as phosphorylation of the C-terminal tail might also be part of the activation process, but the full activation pathway is largely unknown and would require future work.

We considered the autoinhibited conformations as "off-cycle" because we believe that these conformations only occur in the presence of this autoinhibition mechanism in cells. If for example binding partners/protein kinase modifications of ATP8B1 can prevent binding of the N- and C-terminal tails, then the flippase would transition directly from E1P to E2P_{active}, as proposed for non-autoinhibited flippases such as ATP11C or Dnf2. Similar "off-cycle" states have also been reported for the bacterial potassium transporter KdpB where the protein is stuck in a E1P like conformation after phosphorylation of a specific Ser in the A-domain by a kinase (Silberberg et al., 2022. <https://elifesciences.org/articles/80988>).

The reviewer is also right that the entire autoinhibition mechanism is transmitted via TM1-2, which adopts a closed conformation with the lipid entry groove blocked. However, our study reveals that this closed conformation exists in equilibrium with an open conformation, which can interact with a transport substrate, but not occlude it due to the autoinhibitory blockade of the A-domain rotation.

Minor point:

The text says (line 78) "Unique to P4-ATPases, the lipid transport is linked to the dephosphorylation reaction..." Isn't this like the K⁺ ions in the Na/K ATPase (they are released upon dephosphorylation)?

Corrected

On page 4, it might be helpful to clarify that the CDC50A extracellular domain is required for forming a functional complex with and chaperoning phospholipid flippases to the plasma membrane [Segawa et al, 2018, Cell Biol 293(6), 2172].

This information has been now added to the introduction of the revised manuscript.

The text says (line 112-113) "We also report that the autoinhibited ATP8B1-CDC50A complex oscillates between an outward-open and ..." I recommend to replace the word oscillates by fluctuates.

Corrected

REVIEWERS' COMMENTS

Reviewer #1 (Remarks to the Author):

The authors have addressed all my comments. I have no further issues with this work.

Reviewer #2 (Remarks to the Author):

In this revision, the authors made several textual and figure changes, most of which I found acceptable. I have only some minor points to add.

1. Generally speaking, many paragraphs in the main text are too long. An extreme case is the section "PI(3,4,5)P2 binding site", which is a single paragraph over more than 2 pages. I strongly advise the authors to break those long paragraphs (more than 5 or 6 sentences) to increase readability.
2. Line 104: the authors should refrain from an unverifiable claim like "unprecedentedly".
3. Figure 1: the right two panels in the top row have red circles, which are not explained in the figure legend.
4. Line 162: I suggest the authors add a sentence describing the differences between the two structures immediately after this sentence and define the "open" and "closed" states here rather than later.
5. Supplementary Figure 13, panel b: there are three red colors, tones of which are only slightly different from each other and thus difficult for readers to distinguish. Different colors should be used.
6. Supplementary Figure 14, top panel: Indicate this structure is from the current study (with a PDB code).
7. Line 211: "indicating stable binding" is an overstatement, given that the simulations were 600ns long, which is very short for the time scale for lipid diffusion. "Suggesting" is more appropriate.
8. Lines 280-282: This statement regarding the water network is speculative. The authors should either add a phrase clearly indicating that this is a speculation or remove it.
9. Line 293: As mentioned in the authors' rebuttal, the density for S403 is not unambiguous for the rotamer. The authors should explicitly indicate in the main text that the rotamer of S403 is not unambiguous in the cryo-EM map, and the rotamers shown in the figure simply represent possible favorable configurations from a rotamer library.
10. Lines 325-329: References are missing.

11. Line 912: “detected by Coot” is not correct. I guess these rotamers are “manually assigned” from Coot’s rotamer library. Percents here do not indicate a likelihood in this structure, and thus it can mislead the readers. Either remove them or add a statement clarifying this.

12. Supplementary Figures: many supplementary figures are missing legends and thus would be difficult to be understood by non-expert readers. The authors should add a legend (with subheadings for individual panels) to all supplementary figures.

Reviewer #3 (Remarks to the Author):

Revised paper is great. Responses were thoughtful.

RESPONSE TO REVIEWER #2

Reviewer #2 (Remarks to the Author):

In this revision, the authors made several textual and figure changes, most of which I found acceptable.

We thank the Reviewer for their appreciation of our work.

I have only some minor points to add.

1. Generally speaking, many paragraphs in the main text are too long. An extreme case is the section “PI(3,4,5)P2 binding site”, which is a single paragraph over more than 2 pages. I strongly advise the authors to break those long paragraphs (more than 5 or 6 sentences) to increase readability.

Paragraphs in the Result section have now been added to improve clarity.

2. Line 104: the authors should refrain from an unverifiable claim like “unprecedentedly”.

*The text has been modified in accordance.
an unprecedently described water molecule → a water molecule*

3. Figure 1: the right two panels in the top row have red circles, which are not explained in the figure legend.

*The Figure 1 legend has been modified in accordance.
“P-Asp”, referes to the aspartyl-phosphoanhydride... → “P-Asp”, circled in red, referes to the aspartyl-phosphoanhydride...*

4. Line 162: I suggest the authors add a sentence describing the differences between the two structures immediately after this sentence and define the “open” and “closed” states here rather than later.

*Thanks for this comment. The text has been modified to improve clarity.
Surprisingly, they are not identical, especially in the N-terminal part of the transmembrane region (Supplementary Fig. 13a) →
Surprisingly, they are not identical, especially in the N-terminal part of the transmembrane region, where TM1 and TM2 have a different orientation either in a “close” or “open” conformation (Supplementary Fig. 19a).*

5. Supplementary Figure 13, panel b: there are three red colors, tones of which are only slightly different from each other and thus difficult for readers to distinguish. Different colors should be used.

Done

6. Supplementary Figure 14, top panel: Indicate this structure is from the current study (with a PDB code).

Corrected, PDB code is now indicated.

7. Line 211: “indicating stable binding” is an overstatement, given that the simulations were 600ns long, which is very short for the time scale for lipid diffusion. “Suggesting” is more appropriate.

The text has been modified in accordance.

indicating stable binding of the two lipids within the cavity → suggesting stable binding of the two lipids within the cavity

8. Lines 280-282: This statement regarding the water network is speculative. The authors should either add a phrase clearly indicating that this is a speculation or remove it.

The text has been modified in accordance.

The organization of this water network can possibly be part of the triggering event of the allosteric transmission through TM2 that enables the A-domain rotation required for the dephosphorylation event. → One may speculate that the organization of this water network is part of the triggering event of the allosteric transmission through TM2 that enables the A-domain rotation required for the dephosphorylation event.

9. Line 293: As mentioned in the authors’ rebuttal, the density for S403 is not unambiguous for the rotamer. The authors should explicitly indicate in the main text that the rotamer of S403 is not unambiguous in the cryo-EM map, and the rotamers shown in the figure simply represent possible favorable configurations from a rotamer library.

The text has been modified in accordance.

The different rotamers of the S403 now interact with the phosphate or with the sn-2 ester carbonyl more tightly (Figure 7b) → In the E2^o·Pi lipid-occluded conformation, the S403 sidechain is less well defined but the different possible rotamers of the S403 can now interact with the phosphate or with the sn-2 ester carbonyl more tightly (Figure 7b, Supplementary Fig. 18b-d).

10. Lines 325-329: References are missing.

Corrected

11. Line 912: “detected by Coot” is not correct. I guess these rotamers are “manually assigned” from Coot’s rotamer library. Percents here do not indicate a likelihood in this structure, and thus it can mislead the readers. Either remove them or add a statement clarifying this.

The Reviewer is right. The text has been modified in accordance.

The different rotamers of S403 detected by Coot are designated as p, m and t with a likelihood of occurrence of 48%, 29% and 22%, respectively → The different rotamers of S403 from Coot’s rotamer library are designated as p, m and t.

12. Supplementary Figures: many supplementary figures are missing legends and thus would be difficult to be understood by non-expert readers. The authors should add a legend (with subheadings for individual panels) to all supplementary figures.

Additional legends have been added.

- Supplementary Fig. 18 : The comparison of the position of the water molecules observed in P4- and P5-ATPase (ATP8B1 and ATP13A2) is similar to the position observed for cations in P1- and P2-ATPases (CopA, ATP2A1 and ATP1A1).

- Supplementary Fig. 20 : The structure of the autoinhibited P4-ATPases in the E2P state display a close lipid entry groove while active proteins are open and filled with lipid substrate.

- Supplementary Fig. 21 : The comparison of the phosphoinositide binding site observed in other P-type ATPases (Drs2 and ATP13A2) reveals that all phosphoinositides bind a similar region of the protein located in a cavity formed by TM7 and TM10 (and TM5 for P4-ATPases) filled with positively charged residues.